# T2I-CompBench: A Comprehensive Benchmark for Open-world Compositional Text-to-image Generation

**Kaiyi Huang**[1]    **Kaiyue Sun**[1]    **Enze Xie**[2]    **Zhenguo Li**[2]    **Xihui Liu**[1*]

[1] The University of Hong Kong    [2] Huawei Noah's Ark Lab

`{huangky, kaiyue}@connect.hku.hk`

`{xie.enze, li.zhenguo}@huawei.com    xihuiliu@eee.hku.hk`

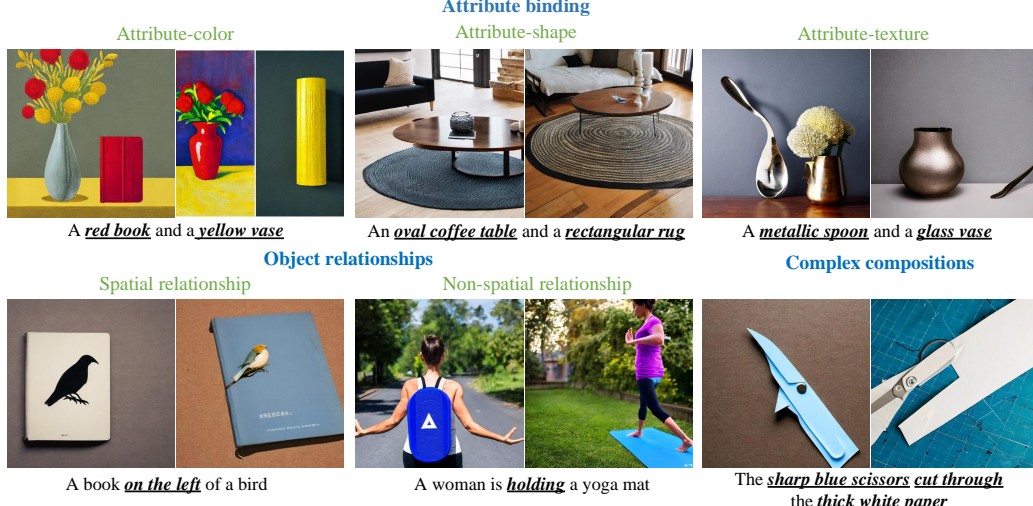

Figure 1: Failure cases of Stable Diffusion v2 [1]. Our compositional text-to-image generation benchmark consists of three categories: attribute binding (including color, shape, and texture), object relationship (including spatial relationship and non-spatial relationship), and complex compositions.

## Abstract

Despite the stunning ability to generate high-quality images by recent text-to-image models, current approaches often struggle to effectively compose objects with different attributes and relationships into a complex and coherent scene. We propose T2I-CompBench, a comprehensive benchmark for open-world compositional text-to-image generation, consisting of 6,000 compositional text prompts from 3 categories (attribute binding, object relationships, and complex compositions) and 6 sub-categories (color binding, shape binding, texture binding, spatial relationships, non-spatial relationships, and complex compositions). We further propose several evaluation metrics specifically designed to evaluate compositional text-to-image generation and explore the potential and limitations of multimodal LLMs for evaluation. We introduce a new approach, Generative mOdel finetuning with Reward-driven Sample selection (GORS), to boost the compositional text-to-image generation abilities of pretrained text-to-image models. Extensive experiments and evaluations are conducted to benchmark previous methods on T2I-CompBench, and to validate the effectiveness of our proposed evaluation metrics and GORS approach. Project page is available at https://karine-h.github.io/T2I-CompBench/.

---

*Corresponding Author

37th Conference on Neural Information Processing Systems (NeurIPS 2023) Track on Datasets and Benchmarks.

# 1 Introduction

Recent progress in text-to-image generation [2, 1, 3, 4, 5, 6] has showcased remarkable capabilities in creating diverse and high-fidelity images based on natural language prompts. However, we observe that even state-of-the-art text-to-image models often fail to compose multiple objects with different attributes and relationships into a complex and coherent scene, as shown in the failure cases of Stable Diffusion [1] in Figure 1. For example, given the text prompt "a blue bench on the left of a green car", the model might bind attributes to the wrong objects or generate the spatial layout incorrectly.

Previous works have explored compositional text-to-image generation from different perspectives, such as concept conjunction [7], attribute binding (focusing on color) [8, 9], and spatial relationship [10]. Most of those works focus on a sub-problem and propose their own benchmarks for evaluating their methods. However, there is no consensus on the problem definition and standard benchmark of compositional text-to-image generation. To this end, we propose a comprehensive benchmark for open-world compositional text-to-image generation, T2I-CompBench. T2I-CompBench consists of three categories and six sub-categories of compositional text prompts: (1) **Attribute binding**. Each text prompt in this category contains at least two objects and two attributes, and the model should bind the attributes with the correct objects to generate the complex scene. This category is divided into three sub-categories (color, shape, and texture) based on the attribute type. (2) **Object relationships**. The text prompts in this category each contain at least two objects with specified relationships between the objects. Based on the type of the relationships, this category consists of two sub-categories, spatial relationship and non-spatial relationship. (3) **Complex compositions**, where the text prompts contain more than two objects or more than two sub-categories mentioned above. For example, a text prompt that describes three objects with their attributes and relationships.

Another challenge is the assessment of compositional text-to-image models. Most previous works evaluate the models by image-text similarity or text-text similarity (between the caption predicted from the generated images and the original text prompts) with CLIPScore [11, 12] or BLIP [13, 14]. However, both metrics do not perform well for compositionality evaluation due to the ambiguity of image captioning and the difficulty of compositional vision-language understanding. To address this challenge, we propose several evaluation metrics for different categories of compositional prompts. We propose *disentangled BLIP-VQA* for attribute binding evaluation to overcome the ambiguous attribute correspondences, *UniDet-based metric* for spatial relationship evaluation, and *3-in-1* metric for complex prompts. We further investigate the potential and limitations of multimodal large language models such as *MiniGPT-4* [15] with Chain-of-Thought [16] for compositionality evaluation.

Finally, we propose a new approach, *Generative mOdel finetuning with Reward-driven Sample selection (GORS)*, for compositional text-to-image generation. We finetune the state-of-the-art Stable Diffusion v2 [1] model with generated images that highly align with the compositional prompts, where the fine-tuning loss is weighted by the reward which is defined as the alignment score between compositional prompts and generated images. This approach is simple but effective in boosting the model's compositional abilities and can serve as a new baseline for future explorations.

In summary, our contributions are three-folded. (1) We propose a comprehensive benchmark for compositional text-to-image generation which consists of 6,000 prompts from 3 categories (attribute binding, object relationships, and complex compositions) and 6 sub-categories (color binding, shape binding, texture binding, spatial relationships, non-spatial relationships, and complex compositions). (2) We propose evaluation metrics that are specifically designed for compositional text-to-image evaluation. Experiments validate that the proposed evaluation metrics are highly correlated with human perception. (3) We benchmark several previous text-to-image models on our proposed benchmark and evaluation metrics, and propose a simple and effective approach, GORS, for boosting the compositionality of text-to-image models.

# 2 Related work

**Text-to-image generation.** Early works [18, 19, 20, 21, 22, 23] explore different network architectures and loss functions based on generative adversarial networks (GAN) [24]. DALL·E [25] achieves impressive results with a transformer and discrete variational autoencoder (VAE) trained on web-scale data. Recently, diffusion models have achieved remarkable success for text-to-image

Table 1: Comparison of compositional text-to-image benchmarks.

| Benchmark | Prompts number and coverage | Vocabulary diversity |
|---|---|---|
| CC-500 [8] | 500 attr bind (color) | 196 nouns, 12 colors |
| ABC-6K [8] | 6,000 attr bind (color) | 3,690 nouns, 33 colors |
| Attn-Exct [9] | 210 attr bind (color) | 24 nouns, 11 colors |
| HRS-comp [17] | 1,000 attr bind (color, size), 2,000 rel (spatial, action) | 620 nouns, 5 colors, 11 spatial, 569 actions |
| T2I-CompBench | 3,000 attr bind (color, shape, texture) 2,000 rel (spatial, non-spatial), 1,000 complex | 2,316 nouns, 33 colors, 32 shapes, 23 textures 7 spatial rel, 875 non-spatial rel |

generation [26, 27, 1, 28, 29]. Current state-of-the-art models such as Stable Diffusion [1] still struggle to compose multiple objects with attributes and relationships in a complex scene. Some recent works attempt to align text-to-image models with human feedback [30, 31]. Concurrent work RAFT [32] proposes reward-ranked fine-tuning to align text-to-image models with certain metrics. Our proposed GORS approach is a simpler finetuning approach that does not require multiple iterations of sample generation and selection.

**Compositional text-to-image generation.** Researchers have delved into various aspects of compositionality in text-to-image generation to achieve visually coherent and semantically consistent results [7, 8, 33, 10]. Previous work focused on concept conjunction and negation [7], attribute binding with colors [8, 9, 34], and spatial relationships between objects [35, 10]. However, those work each target at a sub-problem, and evaluations are conducted in constrained scenarios. Our work is the first to introduce a comprehensive benchmark for compositional text-to-image generation.

**Benchmarks for text-to-image generation.** Early works evaluate text-to-image on CUB birds [36], Oxford flowers [37], and COCO [38] which are easy with limited diversity. As the text-to-image models become stronger, more challenging benchmarks have been introduced. DrawBench [3] consists of 200 prompts to evaluate counting, compositions, conflicting, and writing skills. DALL-EVAL [39] proposes PaintSkills to evaluate visual reasoning skills, image-text alignment, image quality, and social bias by 7,330 prompts. HE-T2I [40] proposes 900 prompts to evaluate counting, shapes, and faces for text-to-image. Several compositional text-to-image benchmarks have also been proposed. Park *et al.* [34] proposes a benchmark on CUB Birds [36] and Oxford Flowers [37] to evaluate the models' ability to generate images with object-color and object-shape compositions. ABC-6K and CC500 [8] benchmarks are proposed to evaluate attribute binding for text-to-image models, but they only focused on color attributes. Concurrent work HRS-Bench [17] is a general-purpose benchmark that evaluates 13 skills with 45,000 prompts. Compositionality is only one of the 13 evaluated skills which is not extensively studied. We propose the first comprehensive benchmark for open-world compositional text-to-image generation, shown in Table 1.

**Evaluation metrics for text-to-image generation.** Existing metrics for text-to-image generation can be categorized into fidelity assessment, alignment assessment, and LLM-based metrics. Traditional metrics such as Inception Score (IS) [41] and Frechet Inception Distance (FID) [42] are commonly used to evaluate the fidelity of synthesized images. To assess the image-text alignment, text-image matching by CLIP [11] and BLIP2 [14] and text-text similarity by BLIP [13] captioning and CLIP text similarity are commonly used. Some concurrent works leverage the strong reasoning abilities of large language models (LLMs) for evaluation [43, 44]. However, there was no comprehensive study on how well those evaluation metrics work for compositional text-to-image generation. We propose evaluation metrics specifically designed for our benchmark and validate that our proposed metrics align better with human perceptions.

## 3 T2I-CompBench

Compositionality of text-to-image models refers to the ability of models to compose different concepts into a complex and coherent scene according to text prompts. It includes composing attributes with objects, composing different objects with specified interactions and spatial relationships, and complex compositions. To provide a clear definition of the problem and to build our benchmark, we introduce three categories and six sub-categories of compositionality, attribute binding (including three sub-categories: color, shape, and texture), object relationships (including two sub-categories: spatial relationship and non-spatial relationship), and complex compositions. We generate 1,000 text prompts (700 for training and 300 for testing) for each sub-category, resulting in 6,000 compositional text

prompts in total. We take the balance between seen *v.s.* unseen compositions in the test set, prompts with fixed sentence template *v.s.* natural prompts, and simple *v.s.* complex prompts into consideration when constructing the benchmark. The text prompts are generated with either predefined rules or ChatGPT [45], so it is easy to scale up. Comparisons between our benchmark and previous benchmarks are shown in Table. 1.

## 3.1 Attribute Binding

A critical challenge for compositional text-to-image generation is attribute binding, where attributes must be associated with corresponding objects in the generated images. We find that models tend to confuse the association between attributes and objects when there are more than one attribute and more than one object in the text prompt. For example, with the text prompt "A room with blue curtains and a yellow chair", the text-to-image model might generate a room with yellow curtains and a blue chair. We introduce three sub-categories, color, shape, and texture, according to the attribute type, and construct 1000 text prompts for each sub-category. For each sub-category, there are 800 prompts with the fixed sentence template "a {adj} {noun} and a {adj} {noun}" (*e.g.*, "a red flower and a yellow vase") and 200 natural prompts without predefined sentence template (*e.g.*, "a room with blue curtains and a yellow chair"). The 300-prompt test set of each sub-category consists of 200 prompts with seen adj-noun compositions (adj-noun compositions appeared in the training set) and 100 prompts with unseen adj-noun compositions (adj-noun compositions not in the training set).

**Color.** Color is the most commonly-used attribute for describing objects in images, and current text-to-image models often confuse the colors of different objects. The 1,000 text prompts related to color binding are constructed with 480 prompts from CC500 [8], 200 prompts from COCO [38], and 320 prompts generated by ChatGPT.

**Shape.** We define a set of shapes that are commonly used for describing objects in images: long, tall, short, big, small, cubic, cylindrical, pyramidal, round, circular, oval, oblong, spherical, triangular, square, rectangular, conical, pentagonal, teardrop, crescent, and diamond. We provide those shape attributes to ChatGPT and ask ChatGPT to generate prompts by composing those attributes with arbitrary objects, for example, "a rectangular clock and a long bench".

**Texture.** Textures are also commonly used to describe the appearance of objects. They can capture the visual properties of objects, such as smoothness, roughness, and granularity. We often use materials to describe the texture, such as wooden, plastic, and rubber. We define several texture attributes and the objects that can be described by each attribute. We generate 800 text prompts by randomly selecting from the possible combinations of two objects each associated with a textural attribute, *e.g.*, "A rubber ball and a plastic bottle". We also generate 200 natural text prompts by ChatGPT.

## 3.2 Object Relationship

When composing objects in a complex scene, the relationship between objects is a critical factor. We introduce 1,000 text prompts for spatial relationships and non-spatial relationships, respectively.

**Spatial relationships.** We use "on the side of", "next to", "near", "on the left of", "on the right of", "on the bottom of", and "on the top of" to define spatial relationships. The two nouns are randomly selected from persons (*e.g.*, man, woman, girl, boy, person, *etc.*), animals (*e.g.*, cat, dog, horse, rabbit, frog, turtle, giraffe, *etc.*), and objects (*e.g.*, table, chair, car, bowl, bag, cup, computer, *etc.*). For spatial relationships including left, right, bottom, and top, we construct contrastive prompts by swapping the two nouns, for example, "a girl on the left of a horse" and "a horse on the left of a girl".

**Non-spatial relationship.** Non-spatial relationships usually describe the interactions between two objects. We prompt ChatGPT to generate text prompts with non-spatial relationships (*e.g.*, "watch", "speak to", "wear", "hold", "have", "look at", "talk to", "play with", "walk with", "stand on", "sit on", *etc.*) and arbitrary nouns.

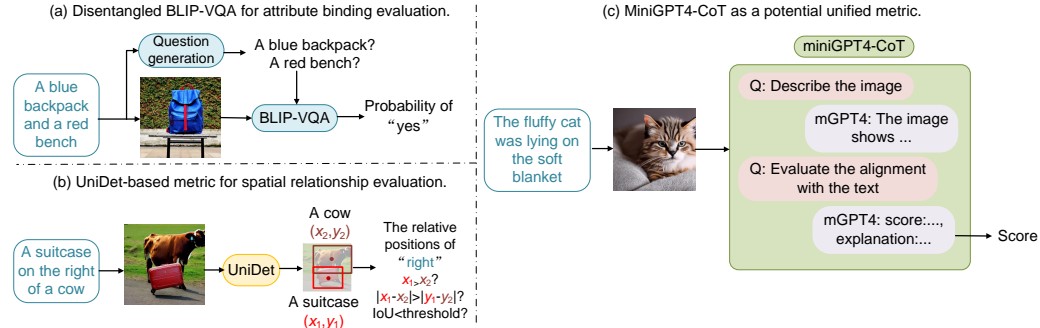

Figure 2: Illustration of our proposed evaluation metrics: (a) Disentangled BLIP-VQA for attribute binding evaluation, (b) UniDet for spatial relationship evaluation, and (c) MiniGPT4-CoT as a potential unified metric.

## 3.3 Complex Compositions

To test text-to-image generation approaches with more natural and challenging compositional prompts in the open world, we introduce 1,000 text prompts with complex compositions of concepts beyond the pre-defined patterns. Regarding the number of objects, we create text prompts with more than two objects, for example, "a room with a blue chair, a black table, and yellow curtains". In terms of the attributes associated with objects, we can use multiple attributes to describe an object (denoted as *multiple attributes*, *e.g.*, "a big, green apple and a tall, wooden table"), or leverage different types of attributes in a text prompt (denoted as *mixed attributes*, *e.g.*, the prompt "a tall tree and a red car" includes both shape and color attributes). We generate 250 text prompts with ChatGPT for each of the four scenarios: *two objects with multiple attributes, two objects with mixed attributes, more than two objects with multiple attributes, and more than two objects with mixed attributes*. Relationship words can be adopted in each scenario to describe the relationships among two or more objects. For each scenario, we split 175 prompts for the training set and 75 prompts for the test set.

## 4 Evaluation Metrics

Evaluating compositional text-to-image generation is challenging as it requires comprehensive and fine-grained cross-modal understanding. Existing evaluation metrics leverage vision-language models trained on large-scale data for evaluation. CLIPScore [11, 12] calculates the cosine similarity between text features and generated-image features extracted by CLIP. Text-text similarity by BLIP-CLIP [9] applies BLIP [13] to generate captions for the generated images, and then calculates the CLIP text-text cosine similarity between the generated captions and text prompts. Those evaluation metrics can measure the coarse text-image similarity, but fails to capture fine-grained text-image correspondences in attribute binding and spatial relationships. To address those limitations, we propose new evaluation metrics for compositional text-to-image generation, shown in Fig. 2. Concretely, we propose *disentangled BLIP-VQA* for attribute binding evaluation, *UniDet-based metric* for spatial relationship evaluation, and *3-in-1* metric for complex prompts. We further investigate the potential and limitations of multimodal large language models such as *MiniGPT-4* [15] with Chain-of-Thought [16] for compositionality evaluation.

### 4.1 Disentangled BLIP-VQA for Attribute Binding Evaluation

We observe that the major limitation of the BLIP-CLIP evaluation is that the BLIP captioning models do not always describe the detailed attributes of each object. For example, the BLIP captioning model might describe an image as "A room with a table, a chair, and curtains", while the text prompt for generating this image is "A room with yellow curtains and a blue chair". So explicitly comparing the text-text similarity might cause ambiguity and confusion.

Therefore, we leverage the visual question answering (VQA) ability of BLIP [13] for evaluating attribute binding. For instance, given the image generated with the text prompt "a green bench

and a red car", we ask two questions separately: "a green bench?", and "a red car?". By explicitly disentangling the complex text prompt into two independent questions where each question contains only one object-attribute pair, we avoid confusion of BLIP-VQA. The BLIP-VQA model takes the generated image and several questions as input and we take the probability of answering "yes" as the score for a question. We compute the overall score by multiplying the probability of answering "yes" for each question. The proposed disentangled BLIP-VQA is applied to evaluate the attribute binding for color, shape, and texture.

## 4.2 UniDet-based Spatial Relationship Evaluation

Most vision-language models perform poorly in reasoning spatial relationships such as "left" and "right". Therefore, we introduce a detection-based spatial relationship evaluation metric. We first use UniDet [46] to detect objects in the generated image. Then we determine the spatial relationship between two objects by comparing the locations of the centers of the two bounding boxes. Denote the center of the two objects as $(x_1, y_1)$ and $(x_2, y_2)$, respectively. The first object is on the left of the second object if $x_1 < x_2, |x_1 - x_2| > |y_1 - y_2|$, and the intersection-over-union (IoU) between the two bounding boxes is below the threshold of $0.1$. Other spatial relationships "right", "top", and "bottom" are evaluated similarly. We evaluate "next to", "near", and "on the side of" by comparing the distances between the centers of two objects with a threshold.

## 4.3 3-in-1 Metric for Complex Compositions Evaluation

Since different evaluation metrics are designed for evaluating different types of compositionality, there is no single metric that works well for all categories. We empirically find that the Disentangled BLIP-VQA works best for attribute binding evaluation, UniDet-baased metric works best for spatial relationship evaluation, and CLIPScore works best for non-spatial relationship evaluation. Thus, we design a 3-in-1 evaluation metric which computes the average score of CLIPScore, Disentangled BLIP-VQA, and UniDet, as the evaluation metric for complex compositions.

## 4.4 Evaluation with Multimodal Large Language Models

By aligning a pretrained visual encoder with a frozen large language model, multimodal large language models such as MiniGPT-4 [15] have demonstrated great abilities in vision-language cross-modal understanding. We leverage MiniGPT-4 with Chain-of-Thought as an evaluation metric by feeding the generated images to the model and asking two questions: "describe the image" and "predict the image-text alignment score". More information on the prompt design is provided in the appendix. We believe more advanced multimodal LLMs have the potential to be a unified evaluation metric in the future, but the current models exhibit limitations such as inaccurate understanding of images and hallucination issues.

# 5   Method

We introduce a simple but effective approach, Generative mOdel finetuning with Reward-driven Sample selection (GORS), to improve the compositional ability of pretrained text-to-image models. Our approach finetunes a pretrained text-to-image model such as Stable Diffusion [1] with generated images that highly align with the compositional prompts, where the fine-tuning loss is weighted by the reward which is defined as the alignment score between compositional prompts and generated images.

Specifically, given the text-to-image model $p_\theta$ and a set of text prompts $y_1, y_2, \cdots, y_n$, we first generate $k$ images for each text prompt, resulting in $kn$ generated images $x_1, x_2, \cdots, x_{kn}$. Text-image alignment scores $s_1, s_2, \cdots, s_{kn}$ are predicted as rewards. We select the generated images whose rewards are higher than a threshold to fine-tune the text-to-image model. The selected set of samples are denoted as $\mathcal{D}_s$. During fine-tuning, we weight the loss with the reward of each sample. Generated images that align with the compositional prompt better are assigned higher loss weights, and vice versa. The loss function for fine-tuning is

$$\mathcal{L}(\theta) = \mathbb{E}_{(x,y,s)\in\mathcal{D}_s}\left[s \cdot \|\epsilon - \epsilon_\theta\left(z_t, t, y\right)\|_2^2\right], \tag{1}$$

Table 2: Benchmarking on attribute binding (color and shape), with scores unnormalized. Blue represents the proposed metric for the category, and green indicates the human evaluation, applicable to the following Table 3 and Table 4.

| Model | Color | | | | | | Shape | | | | | |
|---|---|---|---|---|---|---|---|---|---|---|---|---|
| | CLIP | B-CLIP | B-VQA-n | B-VQA | mGPT-CoT | Human | CLIP | B-CLIP | B-VQA-n | B-VQA | mGPT-CoT | Human |
| Stable v1-4 [1] | 0.3214 | 0.7454 | 0.5875 | 0.3765 | 0.7424 | 0.6533 | 0.3112 | 0.7077 | 0.6771 | 0.3576 | 0.7197 | 0.6160 |
| Stable v2 [1] | 0.3335 | 0.7616 | 0.7249 | 0.5065 | 0.7764 | 0.7747 | **0.3203** | 0.7191 | 0.7517 | 0.4221 | 0.7279 | 0.6587 |
| Composable v2 [7] | 0.3178 | 0.7352 | 0.5744 | 0.4063 | 0.7524 | 0.6187 | 0.3092 | 0.6985 | 0.6125 | 0.3299 | 0.7124 | 0.5133 |
| Structured v2 [8] | 0.3319 | 0.7626 | 0.7184 | 0.4990 | 0.7822 | 0.7867 | 0.3178 | 0.7177 | 0.7500 | 0.4218 | 0.7228 | 0.6413 |
| Attn-Exct v2 [9] | 0.3374 | **0.7810** | 0.8362 | 0.6400 | **0.8194** | 0.8240 | 0.3189 | **0.7209** | 0.7723 | 0.4517 | 0.7299 | 0.6360 |
| GORS-unbaised (ours) | 0.3390 | 0.7667 | 0.8219 | 0.6414 | 0.7987 | 0.8253 | 0.3175 | 0.7149 | 0.7630 | 0.4546 | 0.7263 | 0.6573 |
| GORS (ours) | **0.3395** | 0.7681 | **0.8471** | **0.6603** | 0.8067 | **0.8320** | 0.2973 | 0.7201 | **0.7937** | **0.4785** | **0.7303** | **0.7040** |

where $(x, y, s)$ is the triplet of the image, text prompt, and reward, and $z_t$ represents the latent features of $x$ at timestep $t$. We adopt LoRA [47] for efficient finetuning.

# 6 Experiments

## 6.1 Experimental Setup

**Evaluated models.** We evaluate the performance of 6 text-to-image models on T2I-CompBench. *Stable Diffusion v1-4* and *Stable Diffusion v2* [1] are text-to-image models trained on large amount of image-text pairs. *Composable Diffusion* [7] is designed for conjunction and negation of concepts for pretrained diffusion models. *Structured Diffusion* [8] and *Attend-and-Excite* [9] are designed for attribute binding for pretrained diffusion models. We re-implement those approaches on Stable Diffusion v2 to enable fair comparisons. *GORS* is our proposed approach which finetunes Stable Diffusion v2 with selected samples and their rewards. Since calculating the rewards for GORS with the automatic evaluation metrics can lead to biased results, we also provide alternative reward models (Appendix D.3) which are different from the evaluation metrics, which is denoted as *GORS-unbiased*.

**Implemenrtation details.** Please find the implementation details in the appendix.

## 6.2 Evaluation Metrics

We generate 10 images for each text prompt in T2I-CompBench for automatic evaluation.

**Previous metrics.** *CLIPScore* [11, 12] (denoted as *CLIP*) calculates the cosine similarity between text features and generated-image features extracted by CLIP. *BLIP-CLIP* [9] (denoted as *B-CLIP*) applies BLIP [13] to generate captions for the generated images, and then calculates the CLIP text-text cosine similarity between the generated captions and text prompts. *BLIP-VQA-naive* (denoted as *B-VQA-n*) applies BLIP VQA to ask a single question (e.g., a green bench and a red car?) with the whole prompt.

**Our proposed metrics.** *Disentangled BLIP-VQA* (denoted as *B-VQA*) is our proposed evaluation metric for attribute binding. *UniDet* is our proposed UniDet-based spatial relationship evaluation metric. *3-in-1* computes the average score of CLIPScore, Disentangled BLIP-VQA, and UniDet, as the evaluation metric for complex compositions. *MiniGPT4-Chain-of-Thought* (denoted as *mGPT-CoT*) serves as a potential unified metric for all types of compositional prompts based on multimodal LLM.

**Human evaluation.** For human evaluation of each sub-category, we randomly select 25 prompts and generate 2 images per prompt, resulting in 300 images generated with 150 prompts per model in total. The testing set includes 300 prompts for each sub-category, resulting in 1800 prompts in total. The prompt sampling rate for human evaluation is $8.33\%$. We utilize Amazon Mechanical Turk and ask three workers to score each generated-image-text pair independently based on the image-text alignment.The worker can choose a score from $\{1, 2, 3, 4, 5\}$ and we normalize the scores by dividing them by 5. We then compute the average score across all images and all workers.

Table 3: Benchmarking on attribute binding (texture) and spatial relationship.

| Model | Texture | | | | | | Spatial | | | | |
|---|---|---|---|---|---|---|---|---|---|---|---|
| | CLIP | B-CLIP | B-VQA-n | B-VQA | mGPT-CoT | Human | CLIP | B-CLIP | UniDet | mGPT-CoT | Human |
| Stable v1-4 [1] | 0.3081 | 0.7111 | 0.6173 | 0.4156 | 0.7836 | 0.7227 | 0.3142 | 0.7667 | 0.1246 | 0.8338 | 0.3813 |
| Stable v2 [1] | 0.3185 | 0.7240 | 0.7054 | 0.4922 | 0.7851 | 0.7827 | 0.3206 | 0.7723 | 0.1342 | 0.8367 | 0.3467 |
| Composable v2 [7] | 0.3092 | 0.6995 | 0.5604 | 0.3645 | 0.7588 | 0.6333 | 0.3001 | 0.7409 | 0.0800 | 0.8222 | 0.3080 |
| Structured v2 [8] | 0.3167 | 0.7234 | 0.7007 | 0.4900 | 0.7806 | 0.7760 | 0.3201 | 0.7726 | 0.1386 | 0.8361 | 0.3467 |
| Attn-Exct v2 [9] | 0.3171 | 0.7206 | 0.7830 | 0.5963 | 0.8062 | 0.8400 | 0.3213 | 0.7742 | 0.1455 | 0.8407 | 0.4027 |
| GORS-unbiased (ours) | 0.3216 | 0.7291 | 0.7778 | 0.6025 | 0.7985 | 0.8413 | 0.3237 | 0.7882 | 0.1725 | 0.8241 | 0.4467 |
| GORS (ours) | 0.3233 | 0.7315 | 0.7991 | 0.6287 | 0.8106 | 0.8573 | 0.3242 | 0.7854 | 0.1815 | 0.8362 | 0.4560 |

Table 4: Benchmarking on the non-spatial relationship and complex compositions.

| Model | Non-spatial | | | | Complex | | | | |
|---|---|---|---|---|---|---|---|---|---|
| | CLIP | B-CLIP | mGPT-CoT | Human | CLIP | B-CLIP | 3-in-1 | mGPT-CoT | Human |
| Stable v1-4 [1] | 0.3079 | 0.7565 | 0.8170 | 0.9653 | 0.2876 | 0.6816 | 0.3080 | 0.8075 | 0.8067 |
| Stable v2 [1] | 0.3127 | 0.7609 | 0.8235 | 0.9827 | 0.3096 | 0.6893 | 0.3386 | 0.8094 | 0.8480 |
| Composable v2 [7] | 0.2980 | 0.7038 | 0.7936 | 0.8120 | 0.3014 | 0.6638 | 0.2898 | 0.8083 | 0.7520 |
| Structured v2 [8] | 0.3111 | 0.7614 | 0.8221 | 0.9773 | 0.3084 | 0.6902 | 0.3355 | 0.8076 | 0.8333 |
| Attn-Exct v2 [9] | 0.3109 | 0.7607 | 0.8214 | 0.9533 | 0.2913 | 0.6875 | 0.3401 | 0.8078 | 0.8573 |
| GORS-unbaised (ours) | 0.3158 | 0.7641 | 0.8353 | 0.9534 | 0.3137 | 0.6888 | 0.3470 | 0.8122 | 0.8654 |
| GORS (ours) | 0.3193 | 0.7619 | 0.8172 | 0.9853 | 0.2973 | 0.6841 | 0.3328 | 0.8095 | 0.8680 |

## 6.3 Quantitative and Qualitative Evaluation

The quantitative evaluation results are reported on attribute binding (color), attribute binding (shape) (Table 2), attribute binding (texture), spatial relationship (Table 3), non-spatial relationship, and comprehensive compositions (Table 4), respectively. Qualitative results are shown Figure 3.

**Comparisons across evaluation metrics.** Previous evaluation metrics, CLIP and BLIP-CLIP, predict similar scores for different models and cannot reflect the differences between models. Our proposed metrics, BLIP-VQA for attribute binding, UniDet for spatial relationship, CLIP for non-spatial relationship, and 3-in-1 for complex compositions, highly align with the human evaluation scores.

**Comparisons across text-to-image models.** (1) Stable Diffusion v2 consistently outperforms Stable Diffusion v1-4 in all types of compositional prompts and evaluation metrics. (2) Although Structured Diffusion built upon Stable Diffusion v1-4 shows great performance improvement in attribute binding as reported in Feng *et al.* [8], Structured Diffusion built upon Stable Diffusion v2 only brings slight performance gain upon Stable Diffusion v2. It indicates that boosting the performance upon a better baseline of Stable Diffusion v2 is more challenging. (3) Composable Diffusion built upon Stable Diffusion v2 does not work well. A similar phenomenon was also observed in previous work [9] that Composable Diffusion often generates images containing a mixture of the subjects. In addition, Composable Diffusion was designed for concept conjunctions and negations so it is reasonable that it does not perform well in other compositional scenarios. (4) Attend-and-Excite built upon Stable Diffusion v2 improves the performance in attribute binding. (5) Previous methods Composable Diffusion [7], Structure Diffusion [8] and Attend-and-Excite [9] are designed for concept conjunction or attribute binding, so they do not result in significant improvements in object relationships. (6) Our proposed approach, GORS, outperforms previous approaches across all types of compositional prompts, as demonstrated by the automatic evaluation, human evaluation, and qualitative results. The evaluation results of GORS-unbiased and GORS significantly exceed the baseline Stable v2. Besides, GORS-unbiased achieves on-par performance with GORS, indicating that our proposed approach is insensitive to the reward model used for selecting samples, and that the proposed approach works well as long as high-quality samples are selected.

**Comparisons across compositionality categories.** According to the human evaluation results, spatial relationship is the most challenging sub-category for text-to-image models, and attribute binding (shape) is also challenging. Non-spatial relationship is the easiest sub-category.

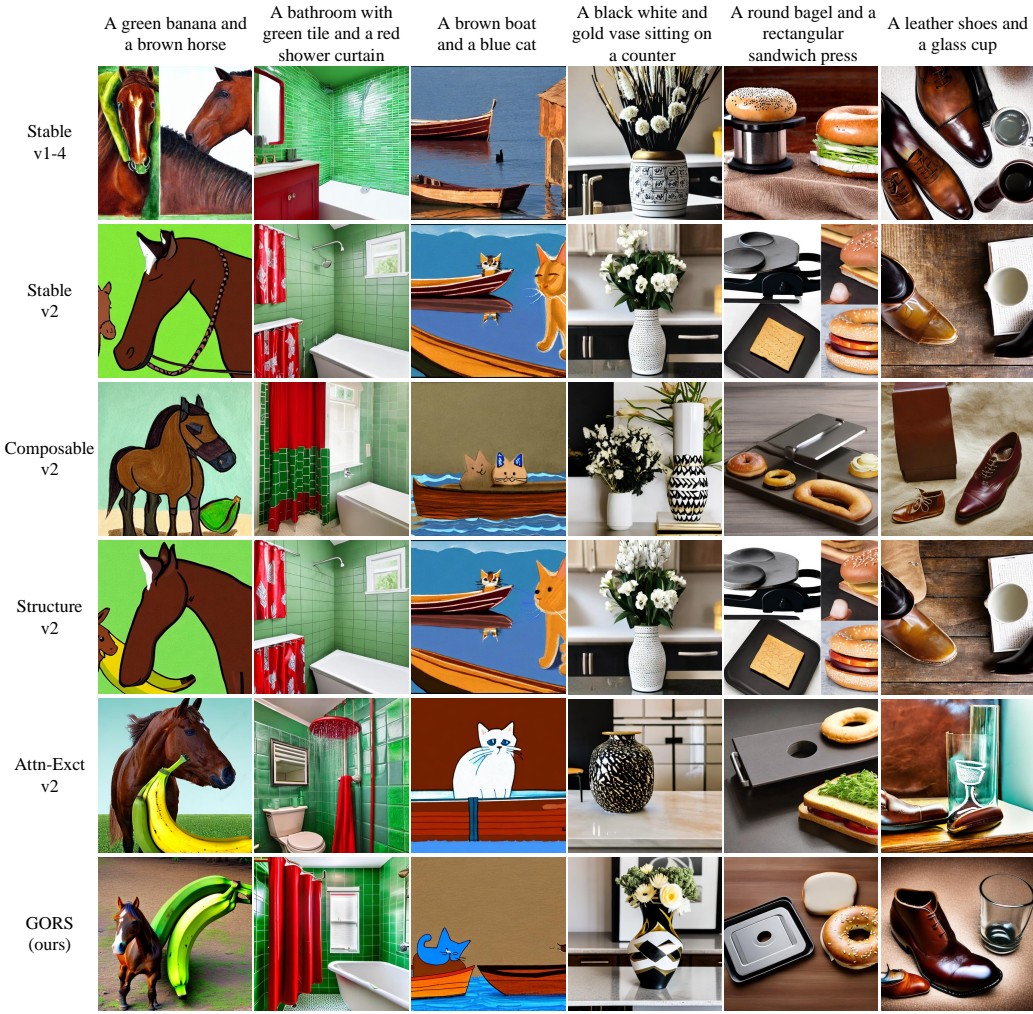

Figure 3: Qualitative comparison between our approach and previous methods.

## 6.4 Human Correlation of the Evaluation Metrics

We calculate Kendall's tau ($\tau$) and Spearman's rho ($\rho$) to evaluate the ranking correlation between automatic evaluation and human evaluation. The scores predicted by each evaluation metric are normalized to 0-1 for better comparison.

The human correlation results are illustrated in Table 5. The results verify the effectiveness of our proposed evaluation metrics, BLIP-VQA for attribute binding, UniDet-based metric for spatial relationships, CLIPScore for non-spatial relationships, and 3-in-1 for complex compositions. MiniGPT4-CoT does not perform well in terms of correlation with human perception, but we believe that the multimodal LLM-based evaluation metrics have the potential to become a unified evaluation metric in the future. We leave explorations on boosting the performance of multimodal LLM for compositional prompts to future work.

## 6.5 Ablation study

We conduct ablation studies with the attribute binding (color) sub-category on our proposed GORS approach and MiniGPT4-CoT evaluation metric.

Table 5: The correlation between automatic evaluation metrics and human evaluation. Our proposed metrics demonstrate a significant improvement over existing metrics in terms of Kendall's $\tau$ and Spearmanr's $\rho$.

| Metric | Attribute-color | | Attribute-shape | | Attribute-texture | | Spatial rel | | Non-spatial rel | | Complex | |
|---|---|---|---|---|---|---|---|---|---|---|---|---|
| | $\tau(\uparrow)$ | $\rho(\uparrow)$ | $\tau(\uparrow)$ | $\rho(\uparrow)$ | $\tau(\uparrow)$ | $\rho(\uparrow)$ | $\tau(\uparrow)$ | $\rho(\uparrow)$ | $\tau(\uparrow)$ | $\rho(\uparrow)$ | $\tau(\uparrow)$ | $\rho(\uparrow)$ |
| CLIP | 0.1938 | 0.2773 | 0.0555 | 0.0821 | 0.2890 | 0.4008 | 0.2741 | 0.3548 | **0.2470** | **0.3161** | 0.0650 | 0.0847 |
| B-CLIP | 0.2674 | 0.3788 | 0.1692 | 0.2413 | 0.2999 | 0.4187 | 0.1983 | 0.2544 | 0.2342 | 0.2964 | 0.1963 | 0.2755 |
| B-VQA-n | 0.4602 | 0.6179 | 0.2280 | 0.3180 | 0.4227 | 0.5830 | – | – | – | – | – | – |
| B-VQA | **0.6297** | **0.7958** | **0.2707** | **0.3795** | **0.5177** | **0.6995** | – | – | – | – | – | – |
| UniDet | – | – | – | – | – | – | 0.4756 | 0.5136 | – | – | – | – |
| 3-in-1 | – | – | – | – | – | – | – | – | – | – | **0.2831** | **0.3853** |
| mGPT | 0.1197 | 0.1616 | 0.1282 | 0.1775 | 0.1061 | 0.1460 | 0.0208 | 0.0229 | 0.1181 | 0.1418 | 0.0066 | 0.0084 |
| mGPT-CoT | 0.3156 | 0.4151 | 0.1300 | 0.1805 | 0.3453 | 0.4664 | 0.1096 | 0.1239 | 0.1944 | 0.2137 | 0.1251 | 0.1463 |
| mGPT-CLIP | 0.2301 | 0.3174 | 0.0695 | 0.0963 | 0.2004 | 0.2784 | 0.1478 | 0.1950 | 0.1507 | 0.1942 | 0.1457 | 0.2014 |

Table 6: Ablation studies on fine-tuning strategy and threshold.

| Metric | FT U-Net only | FT CLIP only | Half threshold | 0 threshold | GORS (ours) |
|---|---|---|---|---|---|
| B-VQA | 0.6216 | 0.5507 | 0.6157 | 0.6130 | **0.6570** |
| mGPT-CoT | 0.7840 | 0.7663 | 0.7886 | 0.7879 | **0.7899** |

**Finetuning strategy.** Our approach finetunes both the CLIP text encoder and the U-Net of Stable Diffusion with LoRA [47]. We investigate the effects of finetuning CLIP only and U-Net only with LoRA. As shown in Table 6, our model which finetunes both CLIP and U-Net performs better.

**Threshold of selecting samples for finetuning.** Our approach fine-tunes Stable Diffusion v2 with the selected samples that align well with the compositional prompts. We manually set a threshold for the alignment score to select samples with higher alignment scores than the threshold for fine-tuning. We experiment with setting the threshold to half of its original value, and setting the threshold to 0 (*i.e.*, use all generated images for finetuning with rewards, without selection). Results in Table 6 demonstrate that half threshold and zero threshold will lead to worse performance.

**MiniGPT4 without Chain-of-Thought.** We compare the evaluation metric of MiniGPT-4 with and without Chain-of-Thought, and with MiniGPT-4 for captioning and CLIP for text-text similarity, denoted as *mGPT-CLIP*. As shown in Table 5, Chain-of-Thought improves the human correlation of MiniGPT4-based evaluation by a large margin.

## 7   Conclusion and Discussions

We propose T2I-CompBench, a comprehensive benchmark for open-world compositional text-to-image generation, consisting of 6,000 prompts from 3 categories and 6 sub-categories. We propose new evaluation metrics and an improved baseline for the benchmark, and validate the effectiveness of the metrics and method by extensive evaluation. One limitation is that we do not have a unified evaluation metric for all categories of the benchmark. Please refer to appendix for failure cases of our proposed metrics. Additionally, our dataset primarily focuses on 2D spatial relationships. We suggest that assessing 3D spatial relationships could integrate depth maps into our existing UniDet-based evaluation metric, which we leave for future work. When studying generative models, researchers need to be aware of the potential negative social impact, for example, it might be abused to generate fake news. We also need to be aware of the bias from the image generators and the evaluation metrics based on pretrained multimodal models. More discussions on the limitation and potential negative social impact are provided in the appendix.

## Acknowledgements

This work is supported in part by HKU Startup Fund, HKU Seed Fund for PI Research - Basic Research, HKU Seed Fund for PI Research - Translational and Applied Research, and HKU FinTech Academy R&D Funding Scheme.

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

# A   Implementation Details

**Evaluation details.** We employ pre-trained models for our proposed evaluation metrics. For BLIP-VQA, we utilize the BLIP w/ ViT-B and CapFilt-L [13] pretrained on image-text pairs and fine-tuned on VQA. We employ the UniDet [46] model Unified_learned_COIM_RS200_6x+2x trained on 4 large-scale detection datasets (COCO [38], Objects365 [48], OpenImages [49], and Mapillary [50]). For CLIPScore, we use the "ViT-B/32" pretrained CLIP model [12, 11]. Finally, for MiniGPT4-CoT, we utilize the Vicuna 13B of MiniGPT4 [15] variant with a temperature setting of 0.7 and a beam size of 1.

**Training details.** We implement our proposed FT-SSWL upon the codebase of diffusers [51] (Apache License), and finetune the self-attention layers of the CLIP text encoder and the attention layers of U-net using LoRA [47]. The model is trained by AdamW optimizer [52] with $\beta_1$=0.9, $\beta_2$=0.999, $\epsilon$=1$e$-8, and weight decay of 0.01. The batch size is 5. The model is trained on 8 32GB NVIDIA v100 GPUs, for 50000-100000 steps.

# B   T2I-CompBench Dataset Construction

This section provides the details of the prompts that ChatGPT uses for generating the text prompts in T2I-CompBench. The text prompts in T2I-CompBench is available at this link.

**Color.** The prompt for ChatGPT is: *Please generate prompts in the format of "a {adj} {noun} and a {adj} {noun}" by using the color adj. , such as "a green bench and a red car".*

**Shape.** (1) For fixed sentence template, the prompt for ChatGPT is: *Please generate prompts in the format of "a {adj} {noun} and a {adj} {noun}" by using the shape adj.: long, tall, short, big, small, cubic, cylindrical, pyramidal, round, circular, oval, oblong, spherical, triangular, square, rectangular, conical, pentagonal, teardrop, crescent, and diamond.* (2) For natural prompts, the prompt for ChatGPT is: *Please generate objects with shape adj. in a natural format by using the shape adj.: long, tall, short, big, small, cubic, cylindrical, pyramidal, round, circular, oval, oblong, spherical, triangular, square, rectangular, conical, pentagonal, teardrop, crescent, and diamond.*

**Texture.** (1) We generate 200 natural text prompts by ChatGPT with the following prompt: *Please generate objects with texture adj. in a natural format by using the texture adj.: rubber, plastic, metallic, wooden, fabric, fluffy, leather, glass.* (2) Besides the ChatGPT-generated text prompts, we also provide the predefined texture attributes and objects that can be described by each texture, as shown in Table 7. We generate 800 text prompts by randomly selecting from the possible combinations of two objects each associated with a textural attribute, *e.g.*, "A rubber ball and a plastic bottle".

Table 7: Textural attributes and associated objects to construct the attribute-texture prompts.

| Textures | Objects |
|---|---|
| Rubber | band, ball, tire, gloves, sole shoes, eraser, boots, mat |
| Plastic | Bottle, bag, toy, cutlery, chair, phone case, container, cup, plate |
| Metallic | car, jewelry, watch, keychain, desk lamp, door knob, spoon, fork, knife, key, ring, necklace, bracelet, earring |
| Wooden | chair, table, picture frame, toy, jewelry box, door, floor, chopsticks, pencils, spoon, knife |
| Fabric | bag, pillow, curtain, shirt, pants, dress, blanket, towel, rug, hat, scarf, sweater, jacket |
| Fluffy | pillow, blanket, teddy bear, rug, sweater, clouds, towel, scarf, hat |
| Leather | jacket, shoes, belt, bag, wallet, gloves, chair, sofa, hat, watch |
| Glass | bottle, vase, window, cup, mirror, jar, table, bowl, plate |

**Non-spatial relation.** The prompt for ChatGPT is: *Please generate natural prompts that contain subjects and objects by using relationship words such as wear, watch, speak, hold, have, run, look at, talk to, jump, play, walk with, stand on, and sit on.*

**Complex.** (1) For 2 objects with mixed attributes, the prompt for ChatGPT is: *Please generate natural compositional phrases, containing 2 objects with each object one adj. from {color, shape, texture} descriptions and spatial (left/right/top/bottom/next to/near/on side of) or non-spatial relationships.* (2) For 2 objects with multiple attributes, the prompt for ChatGPT is: *Please generate natural compositional phrases, containing 2 objects with several adj. from {color, shape, texture} descriptions and spatial (left/right/top/bottom/next to/near/on side of) or non-spatial relationships.* (3) For multiple

objects with mixed attributes, the prompt for ChatGPT is: *Please generate natural compositional phrases, containing multiple objects (number>2) with each one adj. from {color, shape, texture} descriptions and spatial (left/right/top/bottom/next to/near/on side of) non-spatial relationships.* (4) For multiple objects with multiple attributes, the prompt for ChatGPT is: *Please generate natural compositional phrases, containing multiple objects (number>2) with several adj. from {color, shape, texture} descriptions and spatial (left/right/top/bottom/next to/near/on side of) or non-spatial relationships.*

## C   Evaluation Metrics

### C.1   Prompts for MiniGPT4-CoT and MiniGPT4 Evaluation

**MiniGPT4-Chain-of-Thought.** In this part, we detail the prompts used for the MiniGPT4-CoT evaluation metric. For each sub-category, we ask two questions in sequence: "describe the image" and "predict the image-text alignment score". Specifically, Table 8 shows the MiniGPT4-CoT prompts for evaluating attribute binding (color, shape, texture). Table 9, Table 10, and Table 11 demonstrate the prompt templates used for spatial relationships, non-spatial relationships, and complex compositions, respectively.

Table 8: Prompts details for mGPT4-CoT evaluation on attribute binding.

| Describe | You are my assistant to identify any objects and their color (shape, texture) in the image. Briefly describe what it is in the image within 50 words. |
|---|---|
| Predict | According to the image and your previous answer, evaluate if there is {adj.+noun} in the image. Give a score from 0 to 100, according the criteria: 100: there is {noun}, and {noun} is {adj}. 75: there is {noun}, {noun} is mostly {adj}. 20: there is {noun}, but it is not {adj}. 10: no {noun} in the image. Provide your analysis and explanation in JSON format with the following keys: score (e.g., 85), explanation (within 20 words). |

Table 9: Prompts details for mGPT4-CoT evaluation on spatial relationship.

| Describe | You are my assistant to identify objects and their spatial layout in the image. Briefly describe the image within 50 words. |
|---|---|
| Predict | According to the image and your previous answer, evaluate if the text "{xxx}" is correctly portrayed in the image. Give a score from 0 to 100, according the criteria: 100: correct spatial layout in the image for all objects mentioned in the text. 80: basically, spatial layout of objects matches the text. 60: spatial layout not aligned properly with the text. 40: image not aligned properly with the text. 20: image almost irrelevant to the text. Provide your analysis and explanation in JSON format with the following keys: score (e.g., 85), explanation (within 20 words). |

Table 10: Prompts details for mGPT4-CoT evaluation on non-spatial relationship.

| Describe | You are my assistant to identify the actions, events, objects and their relationships in the image. Briefly describe the image within 50 words. |
|---|---|
| Predict | According to the image and your previous answer, evaluate if the text "{xxx}" is correctly portrayed in the image. Give a score from 0 to 100, according the criteria: 100: the image accurately portrayed the actions, events and relationships between objects described in the text. 80: the image portrayed most of the actions, events and relationships but with minor discrepancies. 60: the image depicted some elements, but action relationships between objects are not correct. 40: the image failed to convey the full scope of the text. 20: the image did not depict any actions or events that match the text. Provide your analysis and explanation in JSON format with the following keys: score (e.g., 85), explanation (within 20 words). |

**MiniGPT-4 without Chain-of-Thought** To guide miniGPT4 in addressing specific compositional problems, we utilize predefined prompts that prompt miniGPT4 to provide a score ranging from 0

Table 11: Prompts details for mGPT4-CoT evaluation on complex compositions.

| Describe | You are my assistant to evaluate the correspondence of the image to a given text prompt. Briefly describe the image within 50 words, focus on the objects in the image and their attributes (such as color, shape, texture), spatial layout and action relationships. |
|---|---|
| Predict | According to the image and your previous answer, evaluate how well the image aligns with the text prompt: {xxx}. Give a score from 0 to 100, according the criteria: 100: the image perfectly matches the content of the text prompt, with no discrepancies. 80: the image portrayed most of the actions, events and relationships but with minor discrepancies. 60: the image depicted some elements in the text prompt, but ignored some key parts or details. 40: the image did not depict any actions or events that match the text. 20: the image failed to convey the full scope in the text prompt. Provide your analysis and explanation in JSON format with the following keys: score (e.g., 85), explanation (within 20 words). |

to 100. For attribute binding, we focus on the presence of specific objects and their corresponding attributes. We utilize a prompt template such as *"Is there {object} in the image? Give a score from 0 to 100. If {object} is not present or if {object} is not {color/shape/texture description}, give a lower score."* We leverage this question for each noun phrase in the text and compute the average score. For the spatial relationships, non-spatial relationships, and complex compositions, we employ a more general prompt template such as *"Rate the overall alignment between the image and the text prompt {prompt}. Give a score from 0 to 100."*.

## C.2 Human Evaluation

We conducted human evaluations on Amazon Mechanical Turk (AMT). Specifically, we ask the annotators to rate the alignment between a generated image and the text prompt used to generate the image. Figure 4, 5, 6, 7, 8, 9 show the interfaces for human evaluation over the 6 sub-categories. We randomly sample 25 prompts from each sub-category and each model, and then randomly select 2 images per prompt. In total, we gather $1,800$ text-image pairs for human evaluation experiments. Each image-text pair is rated by 3 human annotators with a score from 1 to 5 according to the image-text alignment. The estimated hourly wage paid to each participant is 9 USD. We spend 270 USD in total on participant compensation.

Text Prompt: **a brown backpack and a blue cow**

Image:

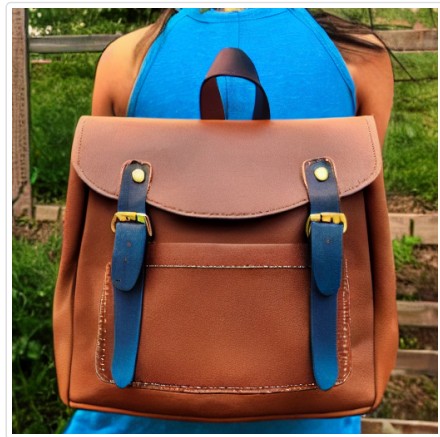

Rate the matching degree of objects' color attributes between the Image and Text Prompt:

○ **5 – Perfect**: all/both objects match their attributes in the text prompt
○ **4 – Good**: basic level of alignment
○ **3 – Not okay**: merely aligned with the text prompt
○ **2 – Bad**: not aligned properly with the text prompt
○ **1 – Poor**: almost irrelevant to the text prompt

Figure 4: AMT Interface for the image-text alignment evaluation on attribute binding (color).

Text Prompt: **an oval sink and a rectangular mirror**

Image:

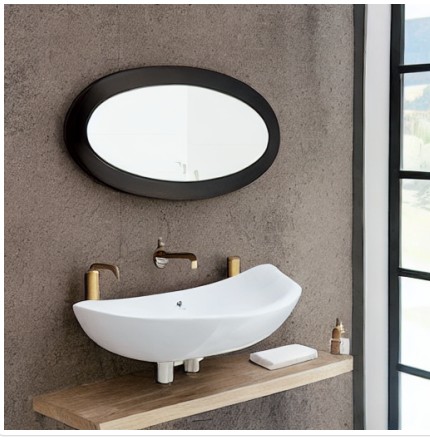

Rate the matching degree of objects' shape attributes between the Image and Text Prompt:

○ **5 – Perfect**: all/both objects match their attributes in the text prompt
○ **4 – Good**: basic level of alignment
○ **3 – Not okay**: merely aligned with the text prompt
○ **2 – Bad**: not aligned properly with the text prompt
○ **1 – Poor**: almost irrelevant to the text prompt

Figure 5: AMT Interface for the image-text alignment evaluation on attribute binding (shape).

Text Prompt: **The glass jar and fluffy ribbon hold the metallic candy on the wooden table**

Image:

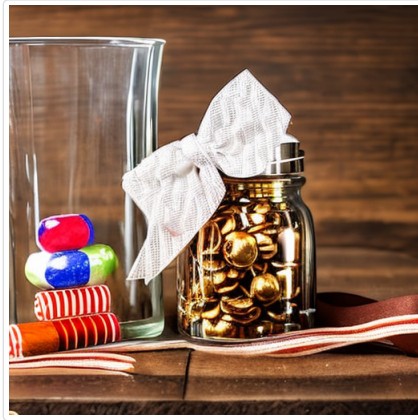

Rate the matching degree of objects' texture attributes between the Image and Text Prompt:

○ **5 – Perfect**: all/both objects match their attributes in the text prompt
○ **4 – Good**: basic level of alignment
○ **3 – Not okay**: merely aligned with the text prompt
○ **2 – Bad**: not aligned properly with the text prompt
○ **1 – Poor**: almost irrelevant to the text prompt

Figure 6: AMT Interface for the image-text alignment evaluation on attribute binding (texture).

Text Prompt: **a vase on the right of a cat**

Image:

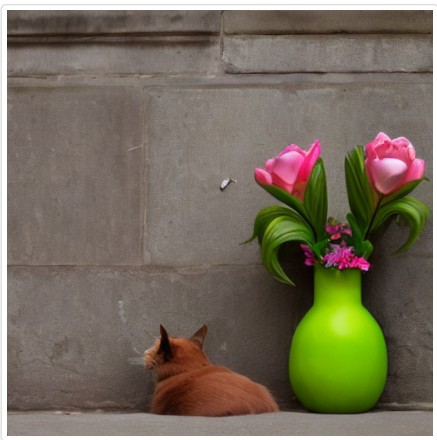

Rate the matching degree of objects' spatial layout between the Image and Text Prompt:

○ **5 – Perfect**: correct spatial layout
○ **4 – Good**: basically correct spatial layout
○ **3 – Not okay**: spatial layout not aligned properly with the text
○ **2 – Bad**: image not aligned properly with the text
○ **1 – Poor**: image almost irrelevant to the text prompt

Figure 7: AMT Interface for the image-text alignment evaluation on spatial relationships.

Text Prompt: **A boat is sailing on a lake**

Image:

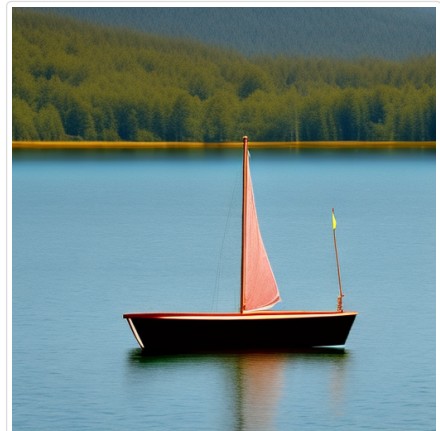

Rate the matching degree of objects' relationship between the Image and Text Prompt:

○ **5 – Perfect**: accurate alignment
○ **4 – Good**: basic level of alignment
○ **3 – Not okay**: action relationship not correct.
○ **2 – Bad**: image not aligned properly with the text
○ **1 – Poor**: image almost irrelevant to the text

Figure 8: AMT Interface for the image-text alignment evaluation on non-spatial relationships.

Text Prompt: **The crisp apple lay beside the rough stone and the silky fabric**

Image:

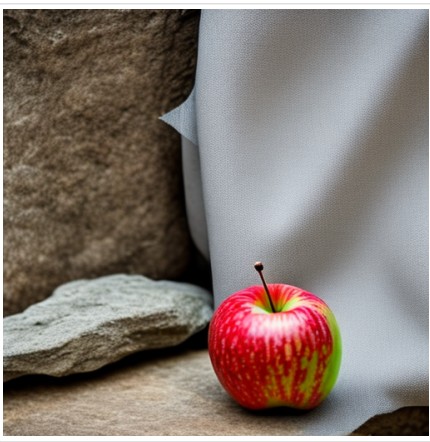

Rate the overall alignment of Image and Text Prompt:

○ **5 – Perfect**: accurate alignment
○ **4 – Good**: basic level of alignment
○ **3 – Not okay**: ignored key parts
○ **2 – Bad**: image not aligned properly with the text
○ **1 – Poor**: image almost irrelevant to the text

Figure 9: AMT Interface for the image-text alignment evaluation on complex compositions.

# D Additional Results

## D.1 Quantitative Results of Seen and Unseen Splits

We provide the seen and unseen splits for the test set, where the unseen set consists of attribute-object pairs that do not appear in the training set. The unseen split tends to include more uncommon attribute-object combinations than seen split. The performance comparison of seen and unseen splits for attribute binding is shown in Table 12. Our observations reveal that our model exhibits slightly lower performance on the unseen set than the seen set.

Table 12: Performances of our model on attribute binding (color, shape, and texture) for seen and unseen sets.

| Metric | Color | | Shape | | Texture | |
|---|---|---|---|---|---|---|
| | Seen | unseen | Seen | unseen | Seen | unseen |
| CLIP | **0.3422** | 0.3283 | 0.2926 | **0.3068** | **0.3240** | 0.3219 |
| B-CLIP | **0.7716** | 0.7612 | **0.7425** | 0.6752 | **0.7569** | 0.6809 |
| B-VQA | **0.7192** | 0.5426 | **0.5500** | 0.3356 | **0.7647** | 0.3567 |
| mGPT | **0.6626** | 0.6780 | **0.6381** | 0.6307 | **0.6773** | 0.6580 |
| mGPT-CoT | **0.8082** | 0.8038 | **0.7510** | 0.6888 | **0.8453** | 0.7412 |

## D.2 MiniGPT-4 Evaluation without Chain-of-Thought

Table 13 shows the additional results of benchmarking on T2I-CompBench of 6 models with MiniGPT-4 without Chain-of-Thought. Results indicate that MiniGPT-4 evaluation without Chain-of-Thought does not strictly align with human evaluation results.

Table 13: mGPT benchmarking on 6 sub-categories in T2I-CompBench.

| Model | Color | | Shape | | Texture | | Spatial | | Non-spatial | | Complex | |
|---|---|---|---|---|---|---|---|---|---|---|---|---|
| | mGPT | Human | mGPT | Human | mGPT | Human | mGPT | Human | mGPT | Human | mGPT | Human |
| Stable v1-4 [1] | 0.6238 | 0.6533 | 0.6130 | 0.6160 | 0.6247 | 0.7227 | 0.8524 | 0.3813 | 0.8507 | 0.9653 | 0.8752 | 0.8067 |
| Stable v2 [1] | 0.6476 | 0.7747 | 0.6154 | 0.6587 | 0.6339 | 0.7827 | 0.8572 | 0.3467 | 0.8644 | 0.9827 | 0.8775 | 0.8480 |
| Composable v2 [7] | 0.6412 | 0.6187 | 0.6153 | 0.5133 | 0.6030 | 0.6333 | 0.8504 | 0.3080 | 0.8806 | 0.8120 | 0.8858 | 0.7520 |
| Structured v2 [8] | 0.6511 | 0.7867 | 0.6198 | 0.6413 | 0.6439 | 0.7760 | 0.8591 | 0.3467 | 0.8607 | 0.9773 | 0.8732 | 0.8333 |
| Attn-Exct v2 [9] | **0.6683** | 0.8240 | 0.6175 | 0.6360 | 0.6482 | 0.8400 | 0.8536 | 0.4027 | 0.8684 | 0.9533 | 0.8725 | 0.8573 |
| GORS-unbiased (ours) | 0.6668 | 0.8253 | 0.6399 | 0.6573 | 0.6389 | 0.8413 | **0.8675** | 0.4467 | 0.8845 | 0.9534 | 0.8876 | 0.8654 |
| GORS (ours) | 0.6677 | **0.8320** | 0.6356 | **0.7040** | **0.6709** | **0.8573** | 0.8584 | **0.4560** | **0.8863** | **0.9853** | **0.8892** | **0.8680** |

## D.3 Reward models to Select Samples for GORS-unbiased

To avoid the bias from selecting samples by evaluation metrics as reward, we introduce new reward models which are different from our proposed evaluation metrics. Specifically, we adopt Grounded-SAM [53] as the reward model for the attribute binding category. We extract the segmentation masks of attributes and their associated nouns separately with Grounded-SAM, and use the Intersection-over-Union (IoU) between the attribute masks and the noun masks together with the grounding mask confidence to represent the attribute binding performance. We apply GLIP-based [54] selection method for spatial relationships. For non-spatial relationships, we adopt BLIP [13] to generate image captions and CLIP [11, 12] to measure the text-text similarity between the generated captions and the input text prompts. For complex compositions, we integrate the 3 aforementioned reward models as the total reward. Those sample selection models are different from the models used as evaluation metrics. The models trained with the new reward models are denoted as GORS-unbiased.

## D.4 Scalability of our proposed approach

To demonstrate the scalability of our proposed approach, we introduce additional 700 prompts of complex compositions to form an extended training set of 1,400 complex prompts. The new prompts are generated with the same methodology as described in the appendix B and they are accessible through this link. We conduct 6 experiments to train the models with different training set sizes, i.e., 25 prompts, 275 prompts, 350 prompts, 700 prompts, 1050 prompts, and 1400 prompts. The results

Table 14: Performances of our model on complex compositons on the 3-in-1 metric

| ours (25) | ours (275) | ours (350) | ours (700) | ours (1050) | ours (1400) |
|-----------|-----------|-----------|-----------|-----------|-----------|
| 0.2596    | 0.3086    | 0.3299    | 0.3328    | 0.3371    | **0.3504** |

in Table 14 show the performance of our model grows with the increase of the training set sizes. The results indicate the potential to achieve better performance by scaling up the training set.

### D.5    Qualitative Results of Ablation Study

We show the qualitative results of the variants in ablation study in Figure 10. When only CLIP is fine-tuned with LoRA, the generated images do not bind attributes to correct objects (for example, Figure 10 Row. 3 Col. 3 and Row. 6 Col. 3). Noticeable improvements are observed in the generated images when U-Net is fine-tuned by LoRA, particularly when both CLIP and U-Net are finetuned together. Furthermore, we delve into the effect of the threshold for selecting images aligned with text prompts for fine-tuning. A higher threshold value enables the selection of images that are highly aligned with text prompts for finetuning, ensuring that only well-aligned examples are incorporated into the finetuning process. In contrast, a lower threshold leads to the inclusion of misaligned images during finetuning, which can degrade the compositional ability of the finetuned text-to-image models (for example, Figure 10 last two columns in Row. 2).

GORS (ours)   FT U-Net only   FT CLIP only   Half threshold   0 threshold

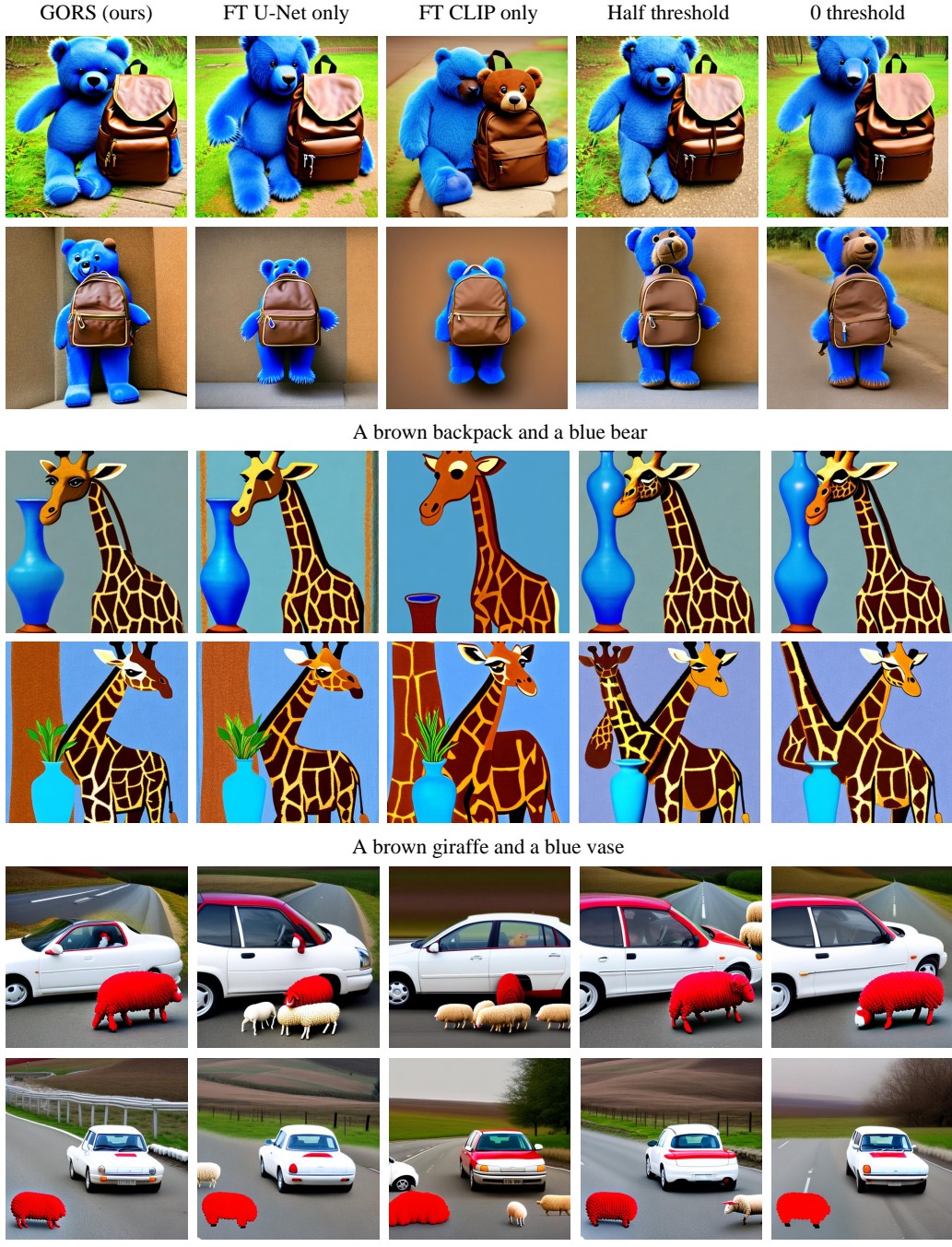

A brown backpack and a blue bear

A brown giraffe and a blue vase

A white car and a red sheep

Figure 10: Qualitative comparison of ablation study on fine-tuning strategy and threshold.

## D.6 Qualitative Results and Comparison with Prior Work

Additional results and comparisons are shown in Figure 11 and Figure 12

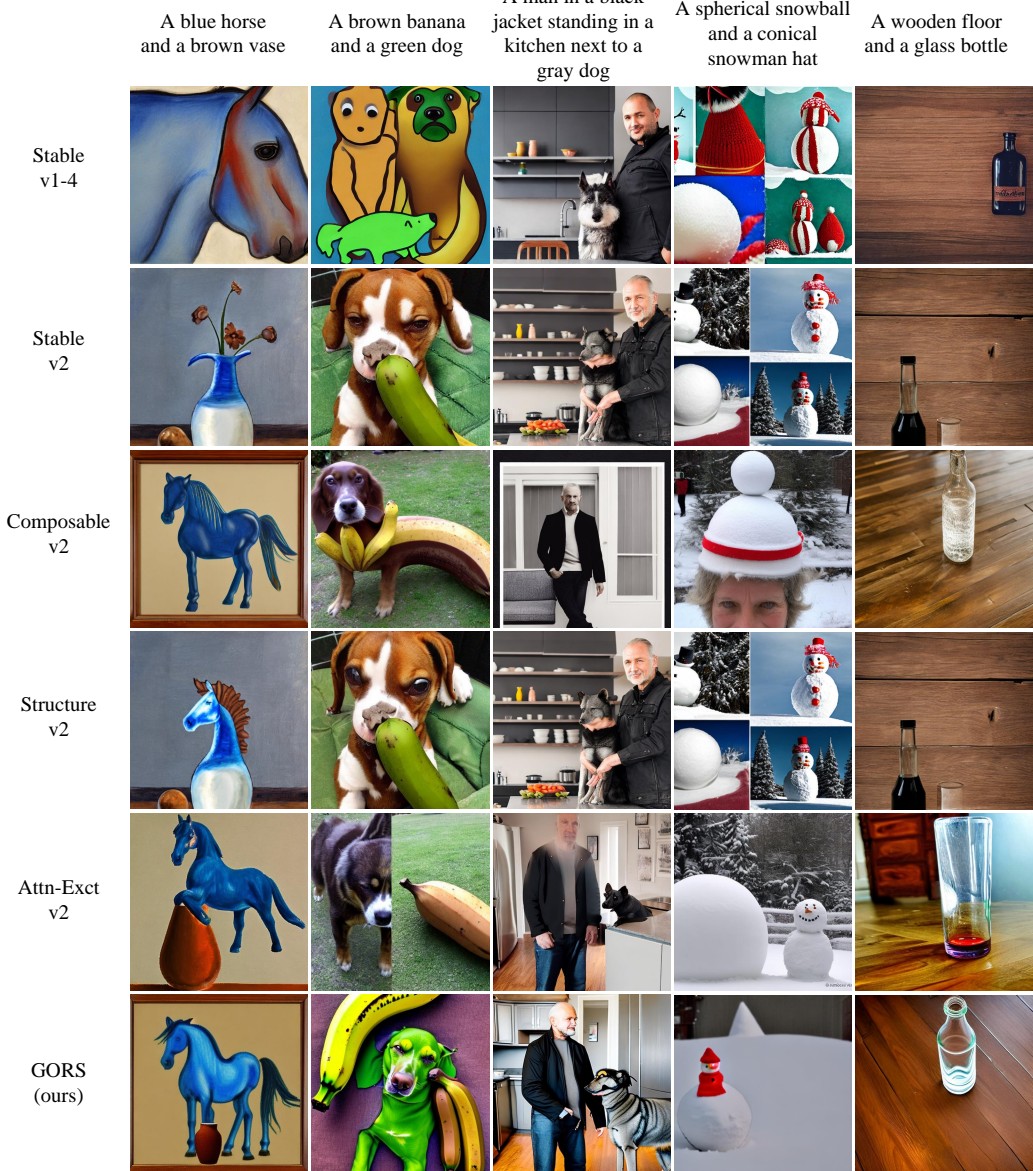

Figure 11: Qualitative comparison between our approach and previous methods.

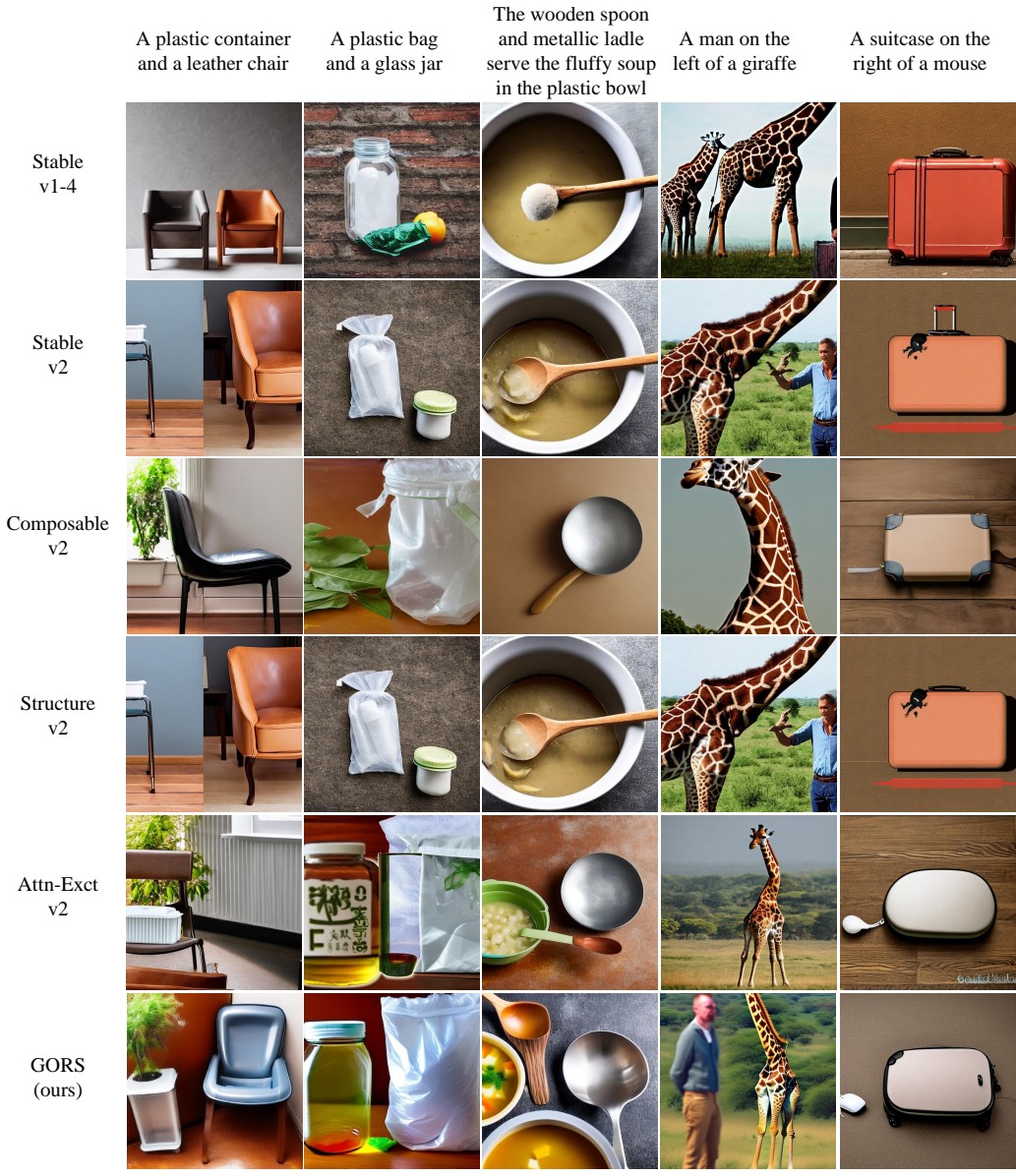

Figure 12: Qualitative comparison between our approach and previous methods.

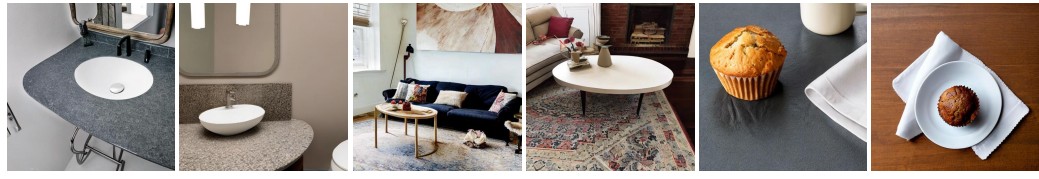

An ***oval sink*** and a ***rectangular mirror***    An ***oval coffee table*** and a ***rectangular rug***    A ***round muffin*** and a ***square napkin***

Figure 13: Failure cases of the evaluation metric BLIP-VQA.

# E   Limitation and Potential Negative Social Impacts

One limitation of our work is the absence of a unified metric for all forms of compositionality. Future research can explore the potential of multimodal LLM to develop a unified metric. Our proposed evaluation metrics are not perfect. As shown by the failure cases in Fig. 13, BLIP-VQA may fail in challenging cases, for example, the objects' shapes are not fully visible in the image, shape's description is uncommon or the objects are not easy to recognize. The UniDet-based evaluation metric is limited to evaluating 2D spatial relationships and we leave 3D spatial relationships for future study. Researchers need to be aware of the potential negative social impact from the abuse of text-to-image models and the biases of hallucinations from image generators as well as pre-trained multimodal models and multimodal LLMs. Future research should exercise caution when working with generated images and LLM-generated content and devise appropriate prompts to mitigate the impact of hallucinations and bias in those models.

