# OpenReview forum: "T2I-CompBench: A Comprehensive Benchmark for Open-world Compositional Text-to-image Generation"
_NeurIPS.cc/2023/Track/Datasets_and_Benchmarks — NeurIPS 2023 Datasets and Benchmarks Poster_

### Official Review · Reviewer_ZzBg · 2023-07-19
**Good contribution of new benchmark and metrics**

**Rating:** 7
**Confidence:** 5
**Clarity:** The paper is well-written and easy to…

**Strengths:**

1. Compared to previous benchmarks (L73-78, Table 1), the paper makes a good contribution by proposing a new benchmark for text-to-image generation, which is more comprehensive (3 categories of compositionality) and making a broader impact by focusing on open-world (instead of domain-specific) generation.
2. The proposed new metrics are reasonable and address the challenges of automated evaluation of compositional text-to-image generation.
3. The paper is well-written and easy to follow.

**Additional Feedback:**

L26: methods.However -> methods. However

L47: missing citation of Chain-of-Thought

L115: missing citation or hyperlink to ChatGPT

**Correctness:**

Overall, the claims made in the submission are correct. However, since the proposed method using the evaluation metrics may skew the results, the claim that the proposed method outperforms previous works may be invalid.

**Documentation:**

The submission provides sufficient details about data collection and reproducing the benchmark results. The license is also included. A URL is provided and accessible.

**Ethics:**

From my perspective, there is no ethical concern in this submission.

**Limitations:**

The paper has an unaddressed limitation—the proposed methods use evaluation metrics, which may skew the results.

**Opportunities For Improvement:**

Although the paper makes a good contribution to the new benchmark and evaluation metrics, the proposed new improved baseline (Section 5) has a huge limitation—using the metrics in the proposed method. As Goodhart’s law states: “When a measure becomes a target, it ceases to be a good measure.” This problem was not only discussed in OOD testing [46] but also highlighted in text-to-image synthesis works [33,22]. For example, [33,22] find the previous methods using DAMSM [20], a model used in the R-Precision metric, skews the R-Precision results. I worry that the proposed method may also overfit the proposed metric.

From my perspective, if the paper does not have the proposed baseline, the submission still
has solid contributions with the new benchmark and metrics and fits very well in the NeurIPS Dataset and Benchmark Track. Therefore, I suggest the paper remove the proposed method and only focus on the first two contributions.

[46] Damien Teney, Kushal Kafle, Robik Shrestha, Ehsan Abbasnejad, Christopher Kanan, and Anton van den Hengel, “On the Value of Out-of-Distribution Testing: An Example of Goodhart’s Law,” in NeurIPS, 2020.

**Relation To Prior Work:**

The paper clearly discusses the difference between this work and previous compositional text-to-image generation benchmarks in Table 1 and L73-78.

**Summary And Contributions:**

The paper proposes T2I-CompBench, a new benchmark to evaluate open-world compositional text-to-image generation. To give a formal definition, the paper introduces three categories of compositionality, i.e., attribute object relationships, and complex compositions, which can further be decomposed into six subcategories. The new dataset contains 6000 text prompts that evaluate the aforementioned three categories of compositionality. To evaluate the compositionality of text-to-image synthesis, the paper proposes three new metrics, i.e., disentangled BLIP-VQA, UniDet-based spatial relationship evaluation, and miniGPT-4 with Chain-of-Thoughts. The paper further proposes a new baseline that uses the proposed metrics to select generated images for finetuning with weighted loss. The experiments show that (1) The proposed new metrics are aligned with human evaluation scores; (2) the proposed new baseline outperforms previous methods.

---

> ### Author Response · Authors · 2023-08-17
> **Response to Official Review by Reviewer ZzBg**
>
> Thank you for your valuable feedback!
>
> > Using the evaluation metrics in the proposed method may skew the results.
>
> Thank you for your feedback and we value this insightful perspective. To address this issue, we **conduct an additional unbiased experiment** where we apply other sample selection methods which are different from evaluation metrics that the models used in our proposed approach. We prove that our approach is effective as long as the selected samples are of high quality, regardless of the model used for selecting the samples.
>
> Specifically, to avoid the bias towards evaluation metrics during sample selection, we adopt Grounded-SAM-based method to select samples for the attribute binding category, GLIP-based selection method for spatial relationships, BLIP-CLIP selection method for non-spatial relationships, and an integration of the 3 selection methods for the complex compositions category. Those sample selection models are different from the models used as evaluation metrics. The details of the new sample selection methods are introduced in Appendix D.3. The evaluation results of our unbiased approach are shown below and updated in Table 2-4 and L#296-300 in the revised paper.
>
> As shown in the results below, the unbiased sample selection approach also leads to improved performance compared with the baseline Stable Diffusion V2, and achieves on-par performance with the previous results which uses biased sample selection methods.
>
> |               | color  |        | shape  |        | texture |        | spatial |        | non-spatial |        | complex |        |
> | ------------- | ------ | ------ | ------ | ------ | ------- | ------ | ------- | ------ | ----------- | ------ | ------- | ------ |
> |               | B-VQA  | Human  | B-VQA  | Human  | B-VQA   | Human  | B-VQA   | Human  | B-VQA       | Human  | B-VQA   | Human  |
> | Stable v2     | 0.5065 | 0.7747 | 0.4221 | 0.6587 | 0.4922  | 0.7827 | 0.1342  | 0.3467 | 0.3127      | 0.9827 | 0.3386  | 0.8480 |
> | GORS-unbiased | 0.6414 | 0.8240 | 0.4546 | 0.6573 | 0.6025  | 0.8413 | 0.1725  | 0.4467 | 0.3158      | 0.9534 | 0.3470  | 0.8654 |
> | GORS          | 0.6603 | 0.8320 | 0.4785 | 0.7040 | 0.6287  | 0.8573 | 0.1815  | 0.4560 | 0.3193      | 0.9853 | 0.3328  | 0.8680 |
>
> In addition, we would like to emphasize that our main contributions are indeed introducing a comprehensive benchmark for compositional text-to-image generation, proposing evaluation metrics specifically for assessing compositional aspects in text-to-image models and evaluating various existing text-to-image models utilizing our proposed benchmark and metrics. The proposed new approach is an additional contribution that would provide references to future research, and we proved that the improvement of the proposed approach is solid with the unbiased sample selection method in the experiment above.
>
>
> >Typos and missing citations.
>
> We have revised the paper according to your suggestions, thank you!

---

> > ### Comment · Reviewer_ZzBg · 2023-08-28
> >
> > I have read other reviewers' comments and the authors' rebuttals. The rebuttal addressed my concerns. I raise my rating to "accept."

---

### Official Review · Reviewer_1VSr · 2023-07-19

**Rating:** 6
**Confidence:** 4
**Clarity:** The authors describe the dataset cons…

**Strengths:**

- The developed dataset is interesting in that it can help evaluate T2I models.
- In addition, the authors also presented a straightforward improvement by fine-tuning with weighted loss.
- The authors conduct a good amount of experiments and ablation studies.
- The paper is well-written and easy to follow.

**Additional Feedback:**

The authors need to improve their dataset website.

**Correctness:**

The evaluation metric may be unreliable. This aspect is unclear in the current draft.

**Documentation:**

The authors provided a link that includes some text prompts for the constructed dataset. However, I found the documents are not fully prepared. It is expected to present a hosting website with detailed documents and source codes for running evaluations. For the current status, it would be difficult for users to access and experiment with the dataset.

**Limitations:**

The authors have discussed the potential negative societal impact.

**Opportunities For Improvement:**

- As the authors use pre-trained models as evaluators, one concern is whether the evaluation results are reliable. MiniGPT4 tends to hallucinate the non-factual objects and MiniGPT4 predictions are not deterministic. As the model outputs can vary over time, simply using MiniGPT4 as an evaluator may not provide reliable evaluation scores.
- The proposed evaluation metrics are simply three existing models. While the authors discussed some variations and specific usages which are interesting, however, the evaluation metrics are running individually. Thus, there are fewer insights to learn from the paper. Putting three models together as evaluators seem like no algorithmic innovation.

**Relation To Prior Work:**

Table 1 presents some of the differences. The authors claim their dataset is the first for open-world compositional T2I, however, the term "open-world" seems to cover very broad topics.

**Summary And Contributions:**

This paper developed a new benchmark called T2I-CompBench. The goal of this benchmark is to evaluate existing T2I models, with a focus on attributes, object relationships, and compositions. For each specific aspect, the authors propose to use a specific model as the evaluator. The authors use BLIP-VQA, UniDet, and miniGPT4 for the 3 evaluation aspects considered, respectively.

---

> ### Author Response · Authors · 2023-08-17
> **Response to Official Review by Reviewer 1VSr**
>
> Thank you for your valuable feedback!
>
> > As the authors use pre-trained models as evaluators, one concern is whether the evaluation results are reliable. MiniGPT4 tends to hallucinate the non-factual objects and MiniGPT4 predictions are not deterministic.
>
> Please refer to the common questions in the general response.
>
> > The proposed evaluation metrics are simply three existing models. While the authors discussed some variations and specific usages which are interesting, however, the evaluation metrics are running individually.
>
> We would like to highlight that our contribution of proposing the evaluation metrics for compositional text-to-image generation is non-trivial and beneficial for the community by setting standard benchmarks and evaluation metrics for future work.
>
> **Although we adopt existing models such as BLIP, we made non-trivial specific adjustments to accommodate our unique requirements and objectives.**
>
> For example, to evaluate attribute binding, we found that applying BLIP directly by simply generating image captions (previous metric BLIP-CLIP) or by asking a single question (e.g., a green bench and a red car?) with the whole prompt (denoted as B-VQA-n) lead to worse evaluation accuracies. We address this issue by disentangling the prompt explicitly and asking multiple questions to the BLIP-VQA model (e.g., a green bench? a red car?). The comparison is shown in the table below and updated in Table 5 in the revised paper. For spatial relationships, the common metrics (BLIP, CLIP) or m-LLM lack the ability to distinguish the spatial relationship, especially left and right. To address the existing limitation, we adopt the object detection tool UniDet and use the relative spatial locations of the detected bounding boxes to determine the spatial relationships.
>
> || Color (τ ρ)    | Shape (τ ρ)    | Texture (τ ρ) |
> | -| -| -| -|
> |B-VQA-n  | 0.4602  0.6179 | 0.2280 0.3180  | 0.4227 0.5830 |
> | B-VQA (ours) | 0.6297  0.7958 | 0.2707  0.3795 | 0.5177 0.6995 |
>
> **Because of the complexity and wide coverage of the compositional benchmark, each evaluation metric has its application scenarios and there are no unified metrics for all cases.**
>
> Our main contribution is that we systematically define the benchmark for compositional text-to-image generation with multiple categories, and explore the most effective evaluation metrics for each category. This comprehensive and systematic categorization allowed us to handle various aspects of compositionality in a structured manner.
>
> The proposed benchmark is comprehensive and covers a large variety of scenarios, and such complex vision-language understanding is beyond the scope of any of the state-of-the-art models. We explore multiple previous and proposed evaluation metrics, but none of them performs well in all scenarios, as demonstrated by the human correlation results in Table 5.
>
> The recently developed multimodal LLMs provide a potential pathway to unified evaluation metrics, but our experiments on MiniGPT-4 indicate that the current multimodal LLMs are not strong enough to be a reliable unified metric. We also investigate other multimodal LLMs including mPlug-Owl[A], MultiModal-GPT[B], and InternChat[C]. Those models also face difficulties in fine-grained understanding of the compositional benchmark, as demonstrated by the failure cases in the image ([link](https://raw.githubusercontent.com/Karine-Huang/test/main/mLLM_attribute.png)).
>
> [A] Ye Q, Xu H, Xu G, et al. mplug-owl: Modularization empowers large language models with multimodality[J]. arXiv preprint arXiv:2304.14178, 2023.
>
> [B] Gong T, Lyu C, Zhang S, et al. Multimodal-gpt: A vision and language model for dialogue with humans[J]. arXiv preprint arXiv:2305.04790, 2023.
>
> [C] Liu Z, He Y, Wang W, et al. Internchat: Solving vision-centric tasks by interacting with chatbots beyond language[J]. arXiv preprint arXiv:2305.05662, 2023.
>
> > The authors claim their dataset is the first for open-world compositional T2I, however, the term "open-world" seems to cover very broad topics.
>
> We use the term “open-world” to emphasize that our dataset does not impose restrictions on the format, style, or content of the input textual prompts or the generated images, resulting in a diverse range of compositional scenarios. In contrast, the previous compositional T2I benchmark [34] was built upon CUB-birds and Oxford-flowers datasets where the domain are restricted to bird and flower images.
> > Documentation, hosting website, and source codes for running evaluations.
>
> We provided the source code with instructions in *Readme.md* in the supplementary materials and the dataset with the URL link. For detailed documents, please find the links of our webpage and Github repository for evaluation usage. Our GitHub repository has garnered substantial attention with 37 stars.
>
> Webpage: https://karine-h.github.io/T2I-CompBench/
>
> Github repo for evaluation code: https://github.com/Karine-Huang/T2I-CompBench

---

> > ### Author Response · Authors · 2023-08-28
> >
> > Dear Reviewer 1VSr,
> >
> > Thanks for your valuable comments and your consideration of our new updates.
> >
> > As the deadline for the author-reviewer discussion phase is approaching (August 29, 01:00 PM PDT), we would like to inquire whether our previous submission has effectively addressed any remaining concerns you might have.
> >
> > Building on your suggestions, we have included extra comparison results and clarified our explanations. All the modifications outlined in our response will be integrated into the revised paper. Should there be any further actions within the discussion period, we intend to address them promptly.
> >
> > We value the time and consideration you've dedicated to our work.
> >
> > Best regards,
> >
> > Submission 369 Authors

---

### Official Review · Reviewer_Qip7 · 2023-07-21

**Rating:** 6
**Confidence:** 3
**Clarity:** Yes

**Strengths:**

The team developed T2I-CompBench, a comprehensive benchmark for evaluating open-world compositional text-to-image synthesis. This benchmark consists of 6,000 compositional text prompts divided into three categories and six sub-categories, focusing on attribute binding, object relationships, and complex compositions. This benchmark is designed to challenge and assess models on more intricate image generation tasks.

New evaluation metrics were proposed to better assess the performance of compositional text-image generation models. These metrics include disentangled BLIP-VQA for attribute binding, a UniDet-based spatial relationship metric, and MiniGPT4-CoT as a unified metric. These metrics aim to overcome the limitations of previous measures that struggled to accurately evaluate compositionality. They are shown some effectiveness through experiments.

The researchers introduced a simple yet effective approach called FT-SSWL. It fine-tunes state-of-the-art text-to-image models, enhancing their ability to generate complex compositional scenes. The FT-SSWL method refines the model using generated images that more align with the compositional prompts. The design of FT-SSWL is pretty similar to hard example mining and makes sense.

Some ablation studies and qualitative comparisons are also included to verify the proposed modules.

**Additional Feedback:**

It seems the supplementary material cannot be unzipped.

**Correctness:**

I generally feel the way to construct dataset is reasonable. So does the design of BLIP-VQA scores.

However, I am unsure if UniDet and miniGPT4 score makes sense. For UniDet, I am unsure how the score can reflect 3D spatial relationship beyond the 2D spatial relationship (e.g., left, top, right). For miniGPT4, I am unsure if the miniGPT4 can produce scores that can model image-text alignment well. If you generate image descriptions via miniGPT4, but use CLIP score to measure, how will it change?

**Documentation:**

It seems some documentation are listed in supplementary while it cannot be unzipped.

**Ethics:**

No serious ethical concerns

**Limitations:**

It is very important to include some failed cases of the proposed evaluation metrics. Then, people can understand what's the limitation of the those metrics. For example, can BLIP-VQA be completely wrong on some combinations? How to determine certain 3D spatial relationships on a 2D image based on the detected bounding box?

Also, the set used for human evaluation is very small. Based on L# 248~249, the author currently only select 25 prompts among a total of 6000 prompts: 25/6000 = 0.4%. The evaluated set is quite small, and I doubt the results still hold for larger-scale sets.

Besides, the proposed metrics seem not be proportional to human evaluation. For example, in Table 3, Structured v2 and Attn-Exct v2 are very different in Human evaluation, but share similar UniDet and mGPT-CoT scores.

Furthermore, according to the results listed in Table 4, the improvements are quite marginally by the proposed FT-SSWL in the non-spatial relationship and complex compositions.

**Opportunities For Improvement:**

Please also refer to the limitation section.

There are several possible points could be improved:
- list more failure cases to provide comprehensive understandings
- enlarge human evaluation scale
- further validate the effectiveness of UniDet and miniGPT4.

**Relation To Prior Work:**

Yes

**Summary And Contributions:**

## After rebuttal

During the rebuttal, the authors addressed my parts of concerns. With more analysis (especially failure cases), this paper in my mind is ok to be accepted. However, I will not fight for its acceptance.

---

This research introduces advancements in the text-to-image synthesis domain by establishing T2I-CompBench, a comprehensive benchmark consisting of 6,000 compositional text prompts across three categories and six sub-categories. In response to existing evaluation limitations, the team proposed new metrics, including disentangled BLIP-VQA, UniDet-based spatial relationship metric, and MiniGPT4-CoT to better gauge the performance of compositional text-image generation models. Furthermore, an fine-tuning approach, Fine-Tuning with Selected Samples and Weighted Loss (FT-SSWL), was developed to enhance the compositionality of text-to-image models. Some experiments are conducted to verify the effectiveness.

---

> ### Author Response · Authors · 2023-08-17
> **Response to Official Review by Reviewer Qip7**
>
> Thank you for your valuable feedback!
> >  Failure cases of the proposed evaluation metrics. BLIP-VQA be wrong on some combinations?
>
> We have added the failure cases of the proposed evaluation metrics in Appendix E. Our proposed BLIP-VQA metric fails when **objects' shapes are not fully visible in the image**.
>
> >  UniDet-based metric for evaluating 3D spatial relationships?
>
> When building our compositional T2I-CompBench, we focus on 2D spatial relationships left/right/top/bottom/next to/besides and do not include 3D spatial relationships front/back, mainly because we found 2D spatial relationships **crucial and more challenging**. Therefore, we design our UniDet-based evaluation metric to focus on **2D spatial relationships** as defined in our T2I-CompBench. We believe 3D spatial relationships can be added to the evaluation metric with an off-the-shelf depth prediction model combined with UniDet, and we leave an in-depth study for future work.
>
> > The set used for human evaluation is very small. Based on L# 248~249, the author currently only select 25 prompts among a total of 6000 prompts: 25/6000 = 0.4%.
>
> We have clarified it in the paper in L# 269-270.
> The 6,000 prompts in T2I-CompBench are divided into training and test set by 7:3 (L# 113-115), resulting in 300 test prompts per sub-category and 1,800 test prompts in total. Among the 1,800 test prompts, we select 25 prompts per sub-category, resulting in 150 prompts in total for human evaluation. Therefore, we use **8.33% (25\*6/(300\*6))** of the test set prompts for human evaluation rather than 0.4%. For each model to be evaluated, we generate 2 images for each of the 150 prompts, and ask 3 users to score each image to reduce bias.
>
> > The proposed metrics seem not be proportional to human evaluation. For example, in Table 3, Structured v2 and Attn-Exct v2 are very different in Human evaluation, but share similar UniDet and mGPT-CoT scores.
>
> Firstly, we would like to emphasize that not all metrics listed in Table 3 are optimal evaluation metrics. Based on their correlation with human evaluation, the best metric for Texture is B-VQA, and for Spatial is UniDet-based metric. mGPT-CoT is an exploration of the potential of mLLMs but we find that the current mLLMs are not good enough to be a unified evaluator for compositionality. Actually, mGPT-CoT shows a low human correlation and we do not regard it as a mature evaluation metric. We've revised the paper to avoid this misunderstanding.
> Secondly, evaluation metrics are not required to be strictly linearly proportional to human evaluation. An evaluation metric can be regarded as a good metric as long as it **provides consistent ranking with human raters**. That's the reason why we adopt Kendall's τ and spearman's ρ rank correlation coefficient to measure the correlation between automatic metrics and human evaluation.
> Thirdly, we apologize for a previous mistake that we put normalized UniDet scores instead of the original scores in Table 3, which might be misleading in terms of the scale of the scores. We have updated Table 3 with the correct scores, and B-VQA scores share a similar trend with the human evaluation for texture, and the UniDet scores for spatial.
>
> > The improvements of the proposed approach are quite marginally in the non-spatial relationship and complex compositions.
>
> The non-spatial relationships are less challenging than other categories, so the improvement space for this categary is not as large as other categories. As we conclude in L# 303, "Non-spatial relationship is the easiest sub-category" for all the 6 sub-catetories.
>
> For complex compositions, we add an ablation study to prove that our proposed approach can be scaled up and perform better with more prompts. We include another GPT-generated 700 prompts in our training set of the complex compositions category. The results show **an improvement of 5.29%**. We have included this experiment in Appendix D.4 and provided the expanded training set in this [link](https://connecthkuhk-my.sharepoint.com/:t:/g/personal/huangky_connect_hku_hk/EZ0PDgNXaiVMiQeytae292sBSMOF0-bnaeilZcg4y17YJA?e=pGjMEs) to facilitate future research.
>
> ||Complex(3-in-1)|
> |-|-|
> | Ours|0.3328|
> | Ours (add training set) | **0.3504** (5.29%↑)|
>
>
>
> > The effectiveness of miniGPT4 as an evaluation metric. If you generate image descriptions via miniGPT4, but use CLIP score to measure, how will it change?
>
> For the effectiveness of MiniGPT-4, please refer to the common questions in the general response.
>
> We test the evaluation metric suggested by the reviewer denoted as mGPT-CLIP. The human correlations in mGPT4-CLIP are significantly lower than the metrics we proposed. Please refer to Table 5 in the updated paper.
>
> > Supplementary cannot be unzipped.
>
> We have uploaded the undated supplementary materials and please try again. If you continue to experience any issues, please let us know, and we'll be more than happy to assist you in resolving them.

---

> > ### Comment · Reviewer_Qip7 · 2023-08-23
> > **Reviewer response**
> >
> > I thank the authors for their effort and for the clarification of the human evaluation. I have further comments below and look forward to your reply.
> >
> >
> > - I am not confident if there is only one reason resulting in failure cases, i.e. objects' shapes are not fully visible in the image. Could you please just generate 1k random examples and visualize via html (or other possible tool)? This way we can inspect the failure case and also check the success/failure ratio.
> >
> > - I personally do not believe 2D spatial relationships is more challenging.
> >
> > - for this claim "our proposed approach can be scaled up and perform better with more prompts", could the author do a series of experiments that training models with different training set sizes and test its performance with all the proposed metrics?

---

> > > ### Author Response · Authors · 2023-08-26
> > > **Response to Official Review by Reviewer Qip7**
> > >
> > > Thank you for your valuable comments!
> > >
> > > > More clarification for failure cases.
> > >
> > > We apologize for the miscommunication. We agree that there are multiple failure modes, and partially visible shapes is one of the typical failure modes but not the only reason for failure cases. To better demonstrate the success and failure cases as well as the success ratio, we generate 1050 examples for visualization in the [link](https://connecthkuhk-my.sharepoint.com/:f:/g/personal/huangky_connect_hku_hk/Ei09aCEYE59Bkv-mWboPQmcBhlGWh-Doh1n1bycKaIxWfA?e=bcWmJk). Specifically, we randomly choose 105 prompts from the attribute binding category, with 35 prompts for each sub-category. For each prompt, we randomly generate 10 images with Stable Diffusion v2, resulting 1050 images in total. We show the text prompt and the automatic evaluation score (BLIP-VQA score) in each image's filename (as the format of "prompt_score.png").
> > > To better visualize the results, we further show some failure cases in this [link](https://karine-huang.github.io/Failure-cases/), and show some prompts, images, and scores that are randomly sampled from the 1050 examples in this [link](https://karine-huang.github.io/sample_demo/) . BLIP-VQA may fail in challenging cases, for example, if the object's shape is not fully visible in the image, shape's description is uncommon (such as "a pentagonal stop sign"), or the objects are not easy to recognize (sometimes even difficult for human to discern).
> > >
> > > We update the discussion of failure cases in the main paper L#336-337 and supplementary materials L#123-125.
> > >
> > >
> > >
> > > > 3D spatial relationships.
> > >
> > > We acknowledge the importance and necessity of understanding 3D spatial relationships for text-to-image generation, but 2D spatial relationships are the common and basic problems we face with current text-to-image generation models. 3D spatial relationships can be evaluated by incorporating depth maps into our UniDet-based evaluation metric. We leave it to be explored by future work. We update this discussion in the main paper L#337-339.
> > >
> > >
> > >
> > > > More experiments on scalability of the model.
> > >
> > > Following the reviewer's suggestion, we conduct a series of experiments to train the models with different training set sizes, i.e., 350 prompts, 700 prompts, 1050 prompts, and 1400 prompts, from the complex compositions category. The 3-in-1 evaluation metric is our only proposed metric for the complex compositions category, because this is the metric with the highest human correlation (shown in main paper Table 5). The results in the table below demonstrates that the performance of our model grows with the increase of the training set sizes, which proves our claim that "our proposed approach can be scaled up and perform better with more prompts". We update the results in the supplementary materials L#103-105 and Table 8.
> > >
> > > |              | 3-in-1     |
> > > | ------------ | ---------- |
> > > | 350 prompts  | 0.3299     |
> > > | 700 prompts  | 0.3328     |
> > > | 1050 prompts | 0.3371     |
> > > | 1400 prompts | **0.3504** |

---

> > > > ### Author Response · Authors · 2023-08-28
> > > >
> > > > Dear Reviewer Qip7,
> > > >
> > > > Thanks for your further comments and your consideration of our new updates.
> > > >
> > > > As the deadline for the author-reviewer discussion phase is approaching (Aug 29, 01:00 PM PDT), we wish to inquire if our earlier post has further addressed your remaining concerns.
> > > >
> > > > In line with your feedback, we have made additional visualizations and have shared these results for the AC’s reference. We will incorporate all the updates mentioned in the rebuttal into the revised paper. We will try to respond further before the discussion period ends, if there's anything more that we may do.
> > > >
> > > > We appreciate your time and consideration.
> > > >
> > > > Best regards,
> > > >
> > > > Submission 369 Authors

---

> > > > ### Comment · Reviewer_Qip7 · 2023-08-28
> > > > **Reviewer response**
> > > >
> > > > I thank the efforts made by the authors.
> > > >
> > > > Re the experiments on scalability of the model: it seems 4 times more prompts results in minor improvements. I wonder if the author could conduct further experiments (in the future revision) with fewer prompts (e.g., 275, 200, 125, 50, 25, 10, 5) to see how the robustness of the proposed model.

---

> > > > > ### Author Response · Authors · 2023-08-29
> > > > >
> > > > > Thank you for your positive feedback!
> > > > > We greatly appreciate your suggestions. In response, we will incorporate your recommendations into the final version of our work. We plan to supplement the revised version with additional experiments as you suggested.

---

### Author Response · Authors · 2023-08-17
**General Response**

We thank all the reviewers for their valuable and insightful comments, and we thank them for the positive feedback.
We are encouraged that they recognize the border impact of the comprehensive benchmark for compositional text-to-image generation (Reviewer Qip7 and Reviewer ZzBg), the effectiveness of our proposed evaluation metrics (Reviewer Qip7 and Reviewer ZzBg), our simple and effective approach  (Reviewer Qip7 and Reviewer 1VSr), and extensive ablation studies and experimental results (eviewer Qip7 and Reviewer 1VSr).

## Common questions

> The effectiveness and reliability of miniGPT4 as an evaluation metric (Reviewer Qip7 and Reviewer 1VSr).

We provide more clarification on MiniGPT-4's evaluation results. Our exploration of MiniGPT-4 was intended to assess the capabilities of multimodal LLM as potential evaluation metrics for compositional text-to-image generation. We explore two prompting ways, namely mGPT and mGPT-CoT to leverage MiniGPT-4's capabilities. We find that careful prompt design can boost its ability, but human evaluation results also show that they are not as good as other proposed metrics (please refer to Table 5). We believe more advanced multimodal LLMs have the potential to be a unified evaluation metric in the future, but the current models are not good enough to solve it because of inaccurate understanding of images and hallucination problems. We revised the paper (L#8-10, L#45-49, L#222-224) to avoid the misunderstandings that MiniGPT4 can be served as a unified evaluation metric.

We have addressed the reviewers’ comments regarding clarity and presentation in all sections. We marked the revised part in red.

## Contributions
We'd like to emphasize our contributions: we are the first to define a comprehensive and systematic benchmark to address the challenges of compositional text-to-image generation. The proposed benchmark covers a wide variety of scenarios, including broader and diverse domains of color, shape, texture, spatial relationships, non-spatial relationships, and complex compositions. Such complex vision-language understanding and generation is beyond the scope of any of the state-of-the-art models. We also propose a set of metrics specifically designed to assess compositionality in text-to-image models. As indicated by Reviewer ZzBg, "*the paper makes a good contribution by proposing a new benchmark for text-to-image generation, which is more comprehensive (3 categories of compositionality) and making a broader impact by focusing on open-world (instead of domain-specific) generation.*" We explore the potential of multimodal Large Language Models as a unified metric for evaluating compositional text-to-image generation, and demonstrate through experimental results that the current multimodal LLMs do not perform well enough to evaluate the compositional text-to-image models. We propose a simple and effective approach to boost the compositional text-to-image generation performance of pre-trained models. Our work not only enriches our comprehension of compositional text-to-image generation but also establishes clear benchmarks and robust evaluation metrics to effectively address compositional challenges. These contributions aim to provide a clearer and more accessible pathway for future research in compositional text-to-image generation.

Thanks to all the reviewers for the appreciation and suggestions of this work, which helped us improve our submission! We have updated the manuscript according to their suggestions and answered their questions in the following.

---

### Decision · Program_Chairs · 2023-09-22

**Decision:**

Accept (Poster)

**Comment:**

All reviewers have agreed that this paper should be accepted after the authors' rebuttal and responses. I concur. The authors have made significant effort to respond to the reviewers' comments and suggestions with broadly satisfactory outcome. The paper makes a good contribution to a rapidly emerging topic of interest with potentially a wider audience of interest.